# The Southern Ocean as the climate's freight train – driving ongoing global warming under zero-emission scenarios with ACCESS-ESM1.5

Matthew A. Chamberlain[a], Tilo Ziehn[b], and Rachel M. Law[b]

[a]CSIRO Environment, Hobart, TAS, Australia
[b]CSIRO Environment, Aspendale, VIC, Australia

**Correspondence:** Matthew A. Chamberlain (matthew.chamberlain@csiro.au)

**Abstract.**

Earth system model experiments presented here explore how the centennial response in the Southern Ocean can drive ongoing global warming even with zero $CO_2$ emissions and declining atmospheric $CO_2$ concentrations. These projections were simulated by the Earth System Model version of the Australian Community Climate and Earth System Simulator (ACCESS-

5 ESM1.5) and motivated by the Zero Emission Commitment Model Intercomparison Project (ZECMIP); ACCESS-ESM1.5 simulated ongoing warming in the ZECMIP experiment that switched or branched to zero emissions after 2000 PgC had been emitted. New experiments presented here each simulated 300 years and included intermediate branch points. In each experiment that branched after emitting more than 1000 PgC, the global climate continues to warm. For the experiment that branched after 2000 PgC, or after 3.5°C of warming from a preindustrial climate, there is 0.37±0.08°C of extra warming after 50 years

of zero emissions, which increases to 0.83±0.08°C after 200 years. All branches show ongoing Southern Ocean warming. The circulation of the Southern Ocean is modified early in the warming climate which contributes to changes in the distribution of both physical and biogeochemical subsurface ocean tracers, such as ongoing warming at intermediate depths and a reduction in deep oxygen south of 60° S.

A simple slab model emulates the global temperatures of the ACCESS-ESM1.5 experiments demonstrating the response here

is primarily due to the slow response of the ocean and the Southern Ocean in particular. Centennial global warming persists when the slab model is forced with $CO_2$ diagnosed from late-branching experiments with other ZECMIP models, confirming the dominant role of ocean physics at these timescales. However, decadal responses changed due to the larger drawdown of $CO_2$ from other models. Slow ongoing warming in the Southern Ocean can be found in ZEC scenarios of most models, though the amplitude and global influence varies.

# 1  Introduction

The Zero-Emission Commitment (ZEC) of the global climate is defined as the warming that would occur after the cessation of anthropogenic $CO_2$ emissions (Matthews and Weaver, 2010). The ZEC is one of the critical terms in the calculation of the remaining carbon emission budget to stay within agreed thresholds of warming (Rogelj et al., 2019); other terms being the amount of historical warming, the transient climate response to ongoing emissions, warming due to non-$CO_2$ greenhouse gases and other forcing agents, and corrections for unaccounted climate feedbacks.

In recent assessments of the remaining carbon budgets for a warming of 1.5 to 2°C, the value of ZEC has typically been assumed to be zero (e.g., Rogelj et al., 2018). The reasoning has been that after the cessation of carbon emissions, the existing carbon in the climate system will redistribute between the atmosphere, land and ocean, reducing the atmospheric $CO_2$ and have a cooling effect on the climate. On the other hand, a slow down in planetary heat uptake, into the ocean in particular, will have a warming effect that has been assumed to cancel the cooling so that there is no net ZEC effect (Rogelj et al., 2019). This assumption had been based largely on results from a limited number of climate simulations. Some previous assessments of the climate under zero-emission scenarios include Gillett et al. (2011) who used the Canadian Earth System Model (ESM) to investigate the centennial responses in ZEC scenarios branching from a contemporary and a future warm climate state. Frölicher et al. (2014) contrasted different multi-centennial responses of two different ESMs. Schwinger et al. (2022) explored the impact of changes in the Atlantic meridional overturning circulation (AMOC) in ZEC and overshoot scenarios with the Norwegian ESM.

In order to reduce the uncertainty associated with the ZEC, a ZEC Model Intercomparison Project (ZECMIP) was designed (Jones et al., 2019) with experiments to simulate climate responses under zero-emission scenarios, branching after varying total emission budgets of $CO_2$. Many of the ESMs that contributed to phase 6 of the Coupled Model Intercomparison Project (CMIP) (Eyring et al., 2016) also participated in ZECMIP, including the ESM version of Australian Community Climate and Earth System Simulator (ACCESS-ESM1.5, Ziehn et al., 2020), as presented in MacDougall et al. (2020). Of the five full ESMs that submitted results for the ZECMIP type-A experiments that branched after emitting 2000 PgC, two ESMs simulated warmer global temperatures 50 years after the cessation of emissions, UKESM1-0-LL (Sellar et al., 2019) and ACCESS-ESM1.5; three ESMs simulated cooling, MIROC-ES2L (Hajima et al., 2020), GFDL-ESM2M (Dunne et al., 2013), and CanESM5 (Swart et al., 2019). The regional responses of ZECMIP models have been assessed by MacDougall et al. (2022) and Cassidy et al. (2023).

In this paper, we investigate the evolution of the climate state and ongoing warming within the ZEC experiments found with the ACCESS-ESM1.5. In the initial results submitted to ZECMIP, ACCESS-ESM1.5 only continued to warm in the 2000 PgC branch. We use extra experiments branching at intermediate points between 1000 and 2000 PgC to evaluate the processes responsible for the ongoing global warming. Section 2 describes the model and summarises the experiments. Section 3 presents the results: time series of global metrics, trajectories of surface temperatures, changes in the ocean circulation and tracer distributions, and, the trajectories of the various ZEC branches with respect to average $CO_2$ and temperature. Section 3 also presents a slab model that emulates the time series of ACCESS-ESM1.5 global average temperatures. The slab is used to

simulate the ACCESS-ESM1.5 response if forced with $CO_2$ from other models, and compare results with other ZECMIP ESMs. Section 4 summarises the work.

## 2 Method

### 2.1 Model Description

The ACCESS-ESM1.5 participated in CMIP6, a global effort to coordinate the design and comparison of climate models and their simulations, and submitted output to several endorsed model intercomparison projects (Mackallah et al., 2022). The model is described in detail by Ziehn et al. (2020). In brief, the atmospheric model is the UK Met Office Unified Model (UM, version 7.3, The HadGEM2 Development Team: et al., 2011) configured at N96 resolution ($1.875°$ longitude and $1.25°$ latitude resolution) with 38 vertical levels, which is coupled to a nominally $1°$ resolution implementation of the Modular Ocean Model (MOM Version 5, Griffies, 2012) and CICE sea ice model (version 4.1, Hunke and Lipscomb, 2010).

Biogeochemical components of the ACCESS-ESM1.5 are Community Atmosphere Biosphere Land Exchange (CABLE, Kowalczyk et al., 2013) and World Ocean Model of Biogeochemistry And Trophic-dynamics (WOMBAT, Oke et al., 2013; Law et al., 2017). CABLE in ACCESS-ESM1.5 is enabled with carbon-nitrogen-phosphorous cycles. The implementation of phosphorous is unique in ACCESS-ESM1.5 and is discussed in Ziehn et al. (2021). WOMBAT is a phosphorous-based nutrient-phytoplankton-zooplankton-detritus model. Both CABLE and WOMBAT include carbon cycles enabling an active carbon cycle in ACCESS-ESM1.5 and the capability to execute these simulations of zero-emission scenarios.

In the development of ACCESS-ESM1.5 from the previous version (ACCESS-ESM1.0, Law et al., 2017) the biases in the physical and biogeochemical states have been reduced and the model has been run and spun up for 1000s of years with prescribed $CO_2$. In the control experiment forced with constant preindustrial conditions (*piControl*, experiment names indicated here with italicises), trends and biases are small in the physical ocean ($-8.5 \times 10^{-5}$ °C century$^{-1}$ in average sea surface temperature) and biogeochemistry (land and ocean carbon fluxes were 0.02 and -0.08 PgC year$^{-1}$ respectively), as reported in Ziehn et al. (2020).

ACCESS-ESM1.5 also executed a control experiment with the interactive carbon cycle enabled (*esm-piControl*) where the atmospheric $CO_2$ was free to evolve, in the same way as in the zero-emission experiments presented here. This *esm-piControl* also benefitted from the long spinup; results available from the Earth System Grid Federation (ESGF, World Climate Research Program, 2023) show that atmospheric $CO_2$ is increasing at $\sim$ 1ppm/100y. Over the first 300 years, the timescale of the experiments presented here, the magnitude of any trends in the global surface air temperatures or sea surface temperature are less than 0.01°C/100y. This stability in the climate state means that model drift is negligible to the results presented.

### 2.2 ZECMIP and supplementary experiments

Experiments under ZECMIP explore idealised zero-emission climate trajectories and are described in detail by Jones et al. (2019). ACCESS-ESM1.5 submitted results for type-A ZECMIP experiments; these experiments branch from *1pctCO2* which

**Table 1.** List of ZECMIP-style experiments with ACCESS-ESM1.5 presented here, including the model year each experiment branches from *1pctCO2*, the anomaly of the 20-year averaged *1pctCO2* global temperature centred at the branch point with respect to *piControl*, and, the change in 20-year averaged temperatures at 25, 50, 100 and 200 years in each ZEC experiment with respect to its branch point. Branches that repeat experiments submitted to ZECMIP are indicated (*); values are from the new experiments presented here that were executed on an updated computer system so that results are equivalent but not identical to results originally submitted to ZECMIP. As a measure of the uncertainty in these values, the standard deviation in the 20-year averaged temperatures from *piControl* is 0.06°C; ZEC values are the differences between two 20-year averages and has an uncertainty of ∼0.08°C.

| Experiment | Carbon emitted (PgC) | Model Year | $\Delta T$ | $ZEC_{25}$ | $ZEC_{50}$ | $ZEC_{100}$ | $ZEC_{200}$ |
|---|---|---|---|---|---|---|---|
| *zec750* | 750* | 53 | 1.39 | 0.05 | -0.06 | -0.10 | -0.10 |
| *zec1000* | 1000* | 67 | 1.82 | 0.02 | 0.01 | 0.02 | -0.02 |
| *zec1250* | 1250 | 80 | 2.20 | 0.14 | 0.31 | 0.17 | 0.20 |
| *zec1500* | 1500 | 93 | 2.61 | 0.28 | 0.34 | 0.44 | 0.53 |
| *zec1750* | 1750 | 104 | 3.10 | 0.24 | 0.34 | 0.47 | 0.58 |
| *zec2000* | 2000* | 115 | 3.51 | 0.33 | 0.37 | 0.57 | 0.83 |

is one of the core CMIP experiments where the climate state warms with a prescribed atmospheric $CO_2$ that increases by 1 percent per year for 140 years. Three ZECMIP-A experiments branch after the emission of 750, 1000 and 2000 PgC, diagnosed from land and ocean carbon fluxes. At these points, emissions are set to zero and the interactive carbon cycle is enabled allowing carbon to exchange freely between climate components, conserving the global carbon content and determining the atmospheric $CO_2$ concentration based on these exchanges. Note, the total cumulative historical $CO_2$ emissions for the period 1850 to 2022

are approaching the lowest branch at 695±70 PgC (Friedlingstein et al., 2023), not accounting for other climate forcing agents. An emission budget of 1000 PgC might be achieved in a few decades if current global emission levels remain about constant. In ZECMIP experiments, only atmospheric $CO_2$ is modified and all other forcings (e.g., from aerosols and $CH_4$) remain at prescribed preindustrial levels. ZECMIP also included type-B experiments that were not simulated by the ACCESS-ESM1.5, where models run entirely with interactive-carbon cycles, starting from a preindustrial climate state, a prescribed budget of

carbon is emitted over a bell-shaped, 100-year pathway before continuing with zero emissions. ZECMIP projections represent idealised zero-emission simulations that are insightful to climate responses for possible future scenarios.

    The work presented here is based on new versions of experiments based on the original ZECMIP branches from the *1pctCO2*, as well as three extra experiments from intermediate branch points (Table 1) that were designed to understand the transition in the climate response between "low" (∼ 1000 PgC) and "high" (∼ 2000 PgC) ZEC branches, branching after the emission of

1250, 1500 and 1750 PgC. All zero-emission experiments were integrated for ∼ 300 years from the branch point to investigate long-term changes within the climate system that can be obscured by internal variability in shorter integrations. These new experiments were executed with the same configuration as the original branches. Table 1 lists the branch points, the model year of the *1pctCO2* at branching and the average global temperature at the branch point, relative to preindustrial.

Throughout the manuscript, the names of ZECMIP and supplementary experiments include the amount of carbon emitted before branching, with text in lowercase, all italicised (e.g. *zec1000*) Where particular ZEC results are shown, they are the difference between 20-year averages from the parent experiment centred on the branch point and the time from branching indicated by the subscripted value of the non-italicised acronym (e.g. $ZEC_{50}$ will indicate the difference centred at 50 years), as defined in MacDougall et al. (2020).

In ZECMIP experiments, all atmospheric concentrations of non-$CO_2$ "greenhouse gases" and aerosols are held constant, and likewise the land use map is maintained with a preindustrial distribution. These idealised zero-emission experiments are different to plausible climate scenarios for the 21st century in which other gases and aerosols are also varying and influencing the climate, and baring a global cataclysm, such an instantaneous transition to zero carbon emissions is perhaps unlikely. However, the results from ZECMIP experiments are expected to be the same as other plausible future climate stabilisation scenarios of corresponding branch point temperatures, such as those proposed by King et al. (2021). The usefulness of the ZECMIP-style experiments is their relatively straight-forward configuration that can readily be adopted by any model with an active carbon cycle, and even across different generations of CMIP.

## 3   Results and Discussion

### 3.1   Global metrics

Figure 1 shows time series of several globally averaged climate metrics from *1pctCO2* and ZEC branches, along with the *piControl*. The time series of surface air temperatures from each ZEC branch are approximately linear (Fig. 1a). The overall rates of change are -0.035°C/100y in *zec750* and +0.315°C/100y in *zec2000*, and the rates vary evenly across intermediate branches. As expected, atmospheric $CO_2$ concentrations drop in all ZEC branches but remain well above preindustrial values (Fig. 1b). Due to the slow response of the deep ocean and the persistent high atmospheric $CO_2$ values, ocean heat continues to increase in all ZEC branches (Fig. 1c). Most of the energy entering the climate system from the imbalance at the top of the atmosphere (TOA, Fig. 1d) is taken up by the ocean of each experiment. In the case of *piControl*, the non-zero TOA balance is consistent with the offset discussed in Ziehn et al. (2020); this offset was still present after a long spinup that stabilised the climate state and the offset isn't associated with any drift in the model. The response of sea ice areas in the Arctic and Antarctic in the ZEC branches are distinct (Figs. 1e and 1f respectively). The Arctic sea ice area largely follows the changes in average global surface temperature (Fig. 1a). On the other hand, the Antarctic sea ice is largely unresponsive at the start of the *1pctCO2* and even the first 100 years of low ZEC branches. However, the longer integrations of the ZEC experiments presented here show reductions in Antarctic sea ice even in the *zec750* branch, where after 200 years sea ice area is outside the range of variability from the *piControl*. The initial sea ice trajectory of *zec2000* is close to that of *1pctCO2*, indicating that the trajectory of the sea ice here at the *zec2000* branch point is already "locked in" and independent of atmospheric forcing for several decades.

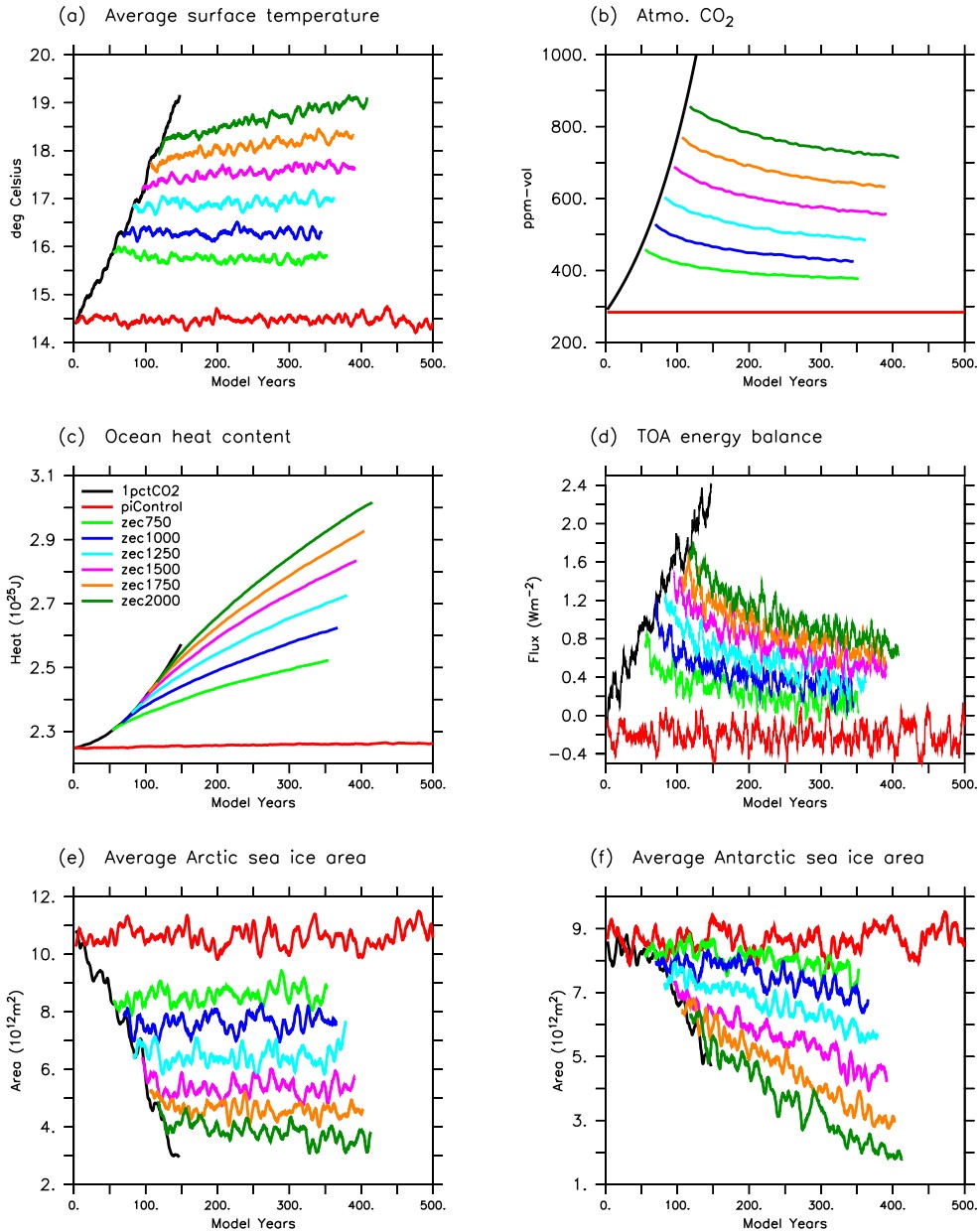

**Figure 1.** Global time series of (a) average near-surface temperature, (b) atmospheric $CO_2$, (c) ocean heat content, (d) top-of-atmosphere energy balance, and sea ice area of the (e) Arctic and (f) Antarctic from the *1pctCO2*, ZEC branches and the *piControl*. Time series are smoothed with 5-year running averages.

## 3.2 Surface temperature changes

Figure 2 shows the evolution in zonally averaged near-surface temperatures in the *1pctCO2* and selected ZECMIP experiments. In *1pctCO2* there is a strong dominant warming in the Arctic responding to the increased climate forcing and global temperatures from higher atmospheric $CO_2$ (Fig. 2a). However, the Arctic and the Northern Hemisphere also cool as the atmospheric $CO_2$ decreases in low ZEC branches (Fig. 2b). The Arctic surface temperature changes appear to follow changes in the global temperature with local amplification from ice albedo feedback. In contrast, the Southern Ocean warms relatively slowly in *1pctCO2* and yet continues to warm in all ZEC branches, consistent with being the region with the greatest inertia in the climate system. For instance, while there is an overall global cooling in *zec750*, after 200 years from branching there is some warming in the same latitude band, 40–65° S, that stands out more clearly in *zec1000* (Fig. 2 b and c). At 65–70° S, the magnitude of warming in low ZEC branches is about the same as warming north of 60° S, whereas these poleward latitudes clearly dominate the warming in *zec2000* corresponding to strong decreases in Antarctic sea ice, and positive feedback on temperature, as seen in Fig. 1f. This slow response of the Southern Ocean has been identified in ocean observations and simulations of the current ocean state by Armour et al. (2016). Here we show the potential influence of this Southern Ocean response to the global climate in zero-emission scenarios.

Figure 3 shows the spatial distribution of the temperature changes of *1pctCO2* and ZEC branch experiments. In low ZEC branches (Fig. 3b and c) there is a broad Southern Ocean response that shows warming across the Atlantic and Indian sectors, extending north to $\sim 45°$ S, though not in the Pacific sector which is about neutral. High ZEC branches (Fig. 3d and e) show larger magnitudes of warming in the Southern Ocean that then drive global changes, note the expanding influences in the zonal time series, Fig. 2d and e. Warming is still evident across the broad regions of the Southern Ocean in high branches, but now the greatest temperature change is located in sea ice regions south of 60° S where changes now trigger positive ice feedbacks.

It is evident that neutral global responses of lowest branches in Fig. 1 obscure significant regional changes. In particular, Fig. 2 and Fig. 3 demonstrate ongoing warming of the order of 1°C after 300 years over the Southern Ocean that is largely compensated by cooling over large continental regions in low ZEC branches. In high ZEC branches, the change in temperature in these continental regions is small with respect to ocean and the Southern Ocean in particular. While there may still be some locations of cooling with decreasing atmospheric $CO_2$ in these high ZEC branches, the cooling is significantly less relative to the cooling in low branches. Also, cooling in these high branches is only found at locations within large continental areas. In contrast, Australia as a smaller continent tends to warm with the neighbouring oceans. Interestingly, one oceanic region shows less warming, and even some cooling in *zec1500* and *zec2000*, around the northern subtropical Pacific which is relatively isolated to warming trends in the Arctic or Southern Ocean.

Similar maps showing the change in surface temperatures over ZEC experiments from other ZECMIP models are presented by MacDougall et al. (2022). Maps of *zec1000* 50-year temperature change between the nine participating ESMs showed significant variability in the regional responses of ZEC simulations. Their Fig. 3 includes results from ACCESS-ESM1.5, however unlike *zec1000* in Fig. 3 here, there is no clear response from the Southern Ocean, whereas some other ZECMIP models indicate strong responses in the North Atlantic, probably associated with changes in the AMOC. As indicated in

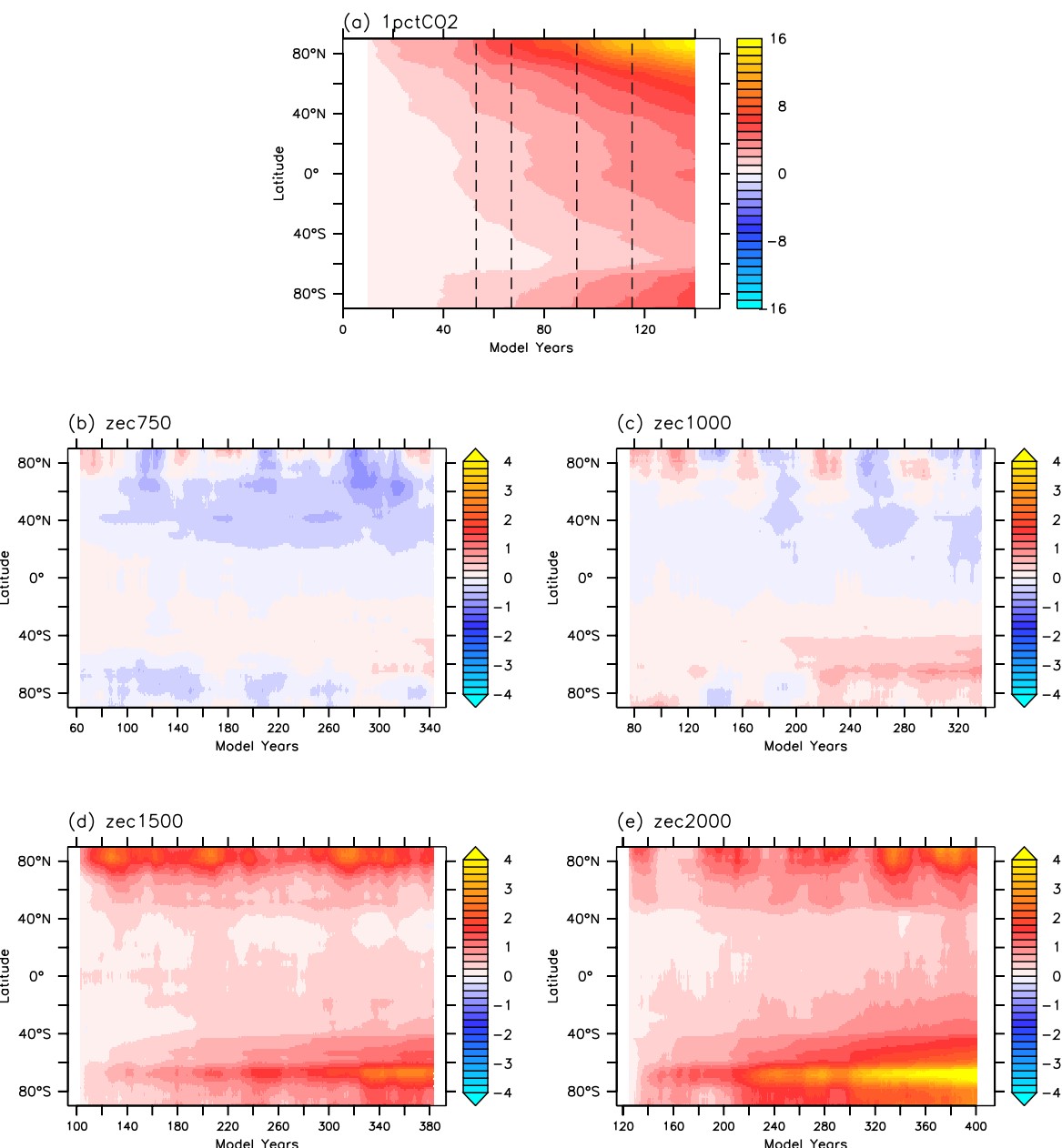

**Figure 2.** Changes in zonally-averaged surface temperatures with time from *1pctCO2* and four of the ZEC branches investigated here. Differences from ZEC branches are with respect to the 20-year averages from *1pctCO2* centered on the branch point, and smoothed with a 20-year filter. Dashed vertical lines in (a) indicate times that the ZEC scenarios shown branch from *1pctCO2*.

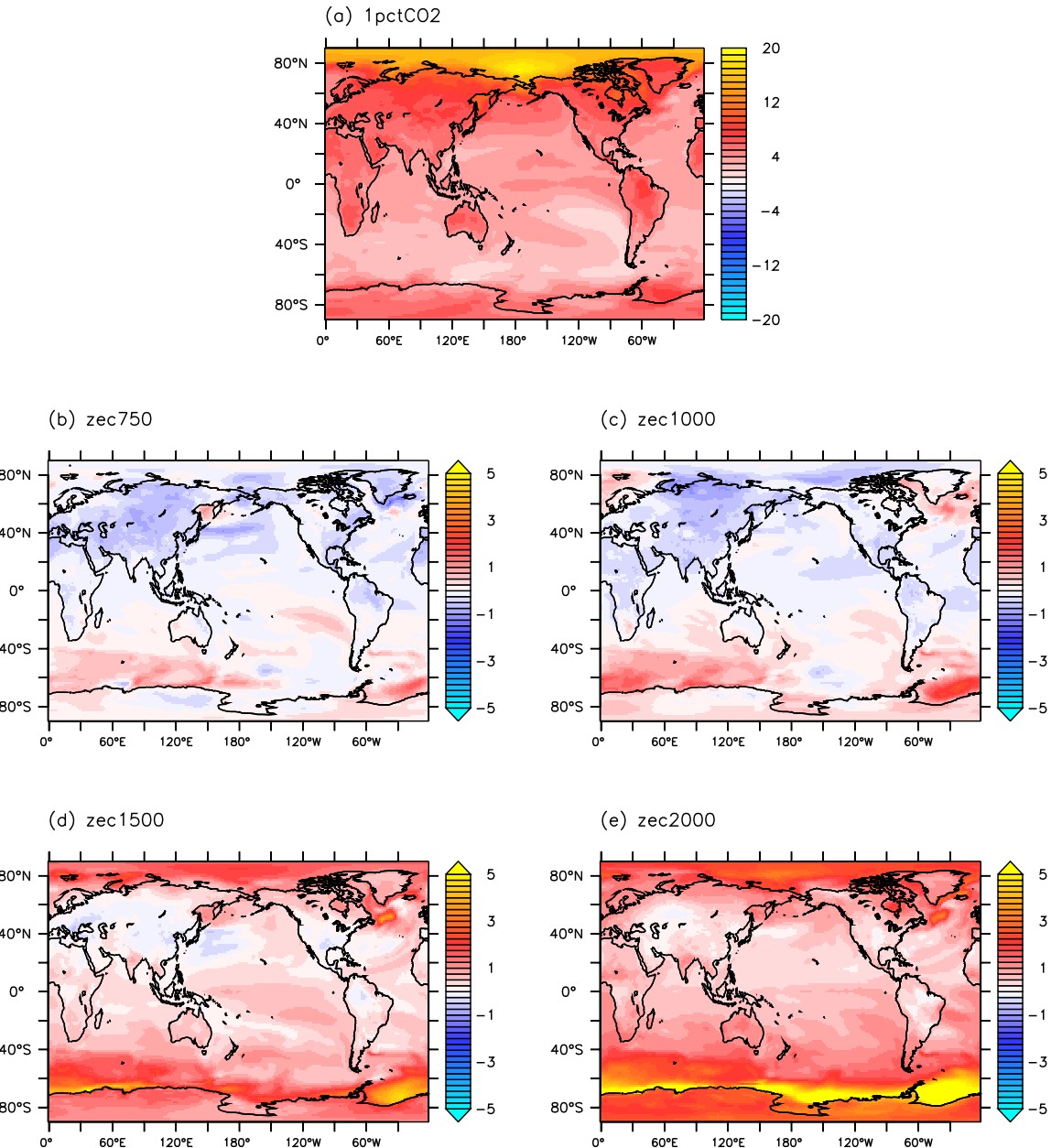

**Figure 3.** Change in surface temperatures. ZEC branch changes are averages of last 20 years with respect to *1pctCO2* centred at the branch point (time difference, $\Delta t=290y$), and *1pctCO2* changes are with respect to *piControl* ($\Delta t=140y$).

Fig. 2 here, the Southern Ocean response in *zec1000* of ACCESS-ESM1.5 does not become apparent until about 200 years after branching and would not appear in the $ZEC_{50}$ results. Also, Gillett et al. (2011) explored zero-emission scenarios with the CanESM1 for centuries after branching from year 2100 of the SRES A2 scenario (Meehl et al., 2007). While the global temperatures were about stable over these centuries, regional temperatures continued to evolve with as much as 3°C of warming over the Southern Ocean, like seen in high ZEC branches of the ACCESS-ESM1.5 here. However, cooling over the northern hemisphere, particularly over high-latitude continental regions and Barents Sea of the Arctic, balanced the southern warming in the global average of CanESM1.

ACCESS-ESM1.5, like many CMIP6 models, doesn't have an active ice sheet component so these simulations will not include the effects of changes in meltwater from ice sheets. Purich and England (2023) show that the inclusion of meltwater around Antarctica in near-future scenarios have a relative cooling effect on surface temperatures across the Southern Ocean (or less warming), due to the reduction in exchange of warmer, deep waters with the surface.

## 3.3 Subsurface changes

### 3.3.1 Overturning

Sections of global meridional overturning streamfunctions from different stages of *1pctCO2* are shown in Fig. 4. Overturning streamfunctions are shown with respect to both depth and density coordinates, and each indicate a decline in strength of circulation of Antarctic Bottom Water. There is a greater influence on the deep bottom water circulation at 3000–4000m in the second 50 years of *1pctCO2* than the first 50 (the change in 4c to 4e being greater than 4a to 4c). The density of the circulation close to the Antarctic coast, south of 60° S, decreases over the course of *1pctCO2* (Fig. 4b, d and f), breaking the coupling to the bottom circulation across the rest of the global ocean. In sections with both coordinates, the Deacon Cell in the Southern Ocean (55° S to 40° S) is stronger and more extensive at the end of *1pctCO2*.

The time series of the overturning shown in the bottom row of Fig. 4 are calculated as the magnitude of the minimum in the streamfunction in depth coordinates at two latitudes, 72° S and 66° S. Variability is high in these overturning values, but with 10-year averaging persistent changes in all ZEC branches become evident, exceeding the significant decadal variability. Even low ZEC branches demonstrate that the small perturbations in average overturning relative to *piControl* do not recover in the 300-year integrations of the branches. Overturning time series in high ZEC branches at 72° S (Fig. 4g) continue to evolve after branching from *1pctCO2* indicating the slow response of deep overturning to changes in surface boundary conditions.

The circulation time series at 66° S appears to collapse as calculated in depth coordinates in Fig. 4h. Average overturning at this position in *piControl* is ∼4 Sv, albeit with significant decadal variability with a range that is also ∼4 Sv, and drops to ∼1 Sv in the low ZEC branches and even < 0.5 Sv in high branches, with no indication of any recovery in the 300-year integrations. However, overturning streamfunctions in density coordinates indicate circulation is ongoing at these latitudes. The timing of the branching of the lowest ZEC branches is about the time that the cell south of 60° S starts to shift to light densities, as seen in Fig 4f.

Not having an active ice sheet component, ACCESS-ESM1.5 will not include the effects on circulation from changes in Antarctic meltwater. Li et al. (2023) and Purich and England (2023) show that the inclusion of Antarctic meltwater in near-future scenarios also act to slow down overturning and the formation of Antarctic Bottom Water, indicating the meltwater impacts will enhance the changes in overturning and the ongoing changes in ocean tracers to what is presented here.

We focus here on the changes in the Southern Ocean. However, changes in the North Atlantic and AMOC have been identified as important features in other analyses of ZEC scenario experiments. MacDougall et al. (2022) assessed regional responses in the *zec1000* from ZECMIP models. One of the significant features at 50 years (ZEC$_{50}$) was in the North Atlantic, which was cooler in some of the models assessed. The ACCESS-ESM1.5 was included in this assessment and the Southern Ocean presented here was not prominent. The *zec1000* is a relatively low branch of the ZEC scenarios presented here, and 50 years is shorter than the timescale that the Southern Ocean response here becomes apparent.

Schwinger et al. (2022) also assessed the impact of AMOC changes, testing various ZEC scenarios with the NorESM2 (Seland et al., 2020), assessing the response of multiple high-emission ZEC branches like done here with ACCESS-ESM1.5, and also overshoot scenarios where negative emissions are used to reduce the $CO_2$ and climate forcing. Schwinger et al. (2022) found significant AMOC responses; temperatures cooled as AMOC weakened and temperatures warmed as AMOC recovered and strengthened. Interestingly, the NorESM2 was included in the multi-model analysis in MacDougall et al. (2022), but the NorESM2 response was not so significant there, possibly because the response in Schwinger et al. (2022) is on centennial time scales and didn't have a strong ZEC$_{50}$ response in the *zec1000* shown.

### 3.3.2 Tracer distributions

The changes in the circulation and surface forcing from the increased climate forcing of *1pctCO2* initiate long-term changes in the distribution of subsurface ocean properties that continue even once the climate forcing decreases and stabilises in the ZECMIP experiments. Figures 5, 6 and 7 show changes in zonally-averaged sections of temperature, salinity and oxygen in the *1pctCO2*, *zec750* and *zec2000*, as well as time series of tracers at selected positions from *1pctCO2*, *piControl* and all ZEC branches. Figure 8 demonstrates the changes in the ocean depth of heat uptake in the different ZEC branches.

The time series (panels j, k, and l of Fig.s 5, 6 and 7) demonstrate that even small changes in circulation and surface forcing of low ZEC branches are sufficient to drive ongoing subsurface changes in heat, salt and oxygen, even if these changes are not expressed at the surface. There is a montonic increase in the rate of change in the subsurface warming with ZEC branches, the fastest warming is in the highest ZEC branches at all positions shown in Fig. 5. Similar responses are seen in oxygen time series (Fig. 7), where high branches generate greater decreases in oxygen. with exceptions that are discussed further below. For the time series of each tracer at 20° S (panel l of each figure), while there are consistent signals across the experiments presented, the magnitudes of low-frequency variability is similar to these signals and a larger 30-year filter is applied to reduce this variability.

Changes in the temperature sections of *1pctCO2* (Fig. 5d and g) are predominately near the surface north of 40° S, with deeper warming near Antarctica down to 2000 m and in the Southern Ocean at 45–50° S down to 1000 m related to a poleward

shift in water masses. In contrast, temperature changes in ZEC branches are predominantly at depth, $\sim$ 500–1500 m north of
60° S and deeper to the south, with less change at the surface.

The uptake of heat in *1pctCO2* and selected ZEC branches are shown in Fig. 8, as changes in the globally averaged temperature with depth and time within each experiment. Consistent with the ocean heat content in Fig. 1c, temperature increases are much larger in high ZEC branches; also, the distribution of temperature increase is shallower in high branches. At the end of the 300 years with zero emissions, the peak temperature increase in *zec2000* is at $\sim$800 m whereas in *zec750* it is at $\sim$1200 m. While temperature still increases below $\sim$200 m in *zec750*, there is some cooling in the upper 100 m in response to the decreasing atmospheric $CO_2$ and reduced climate forcing. In contrast, in *zec2000*, the highest temperature increase is closer to the surface and has a greater influence on the upper ocean and surface.

Figure 6 shows changes in zonal averages of salinity from *1pctCO2* and selected ZEC branches. The changes in zonal salinity in *1pctCO2* vary spatially and are distinct from temperature changes, with freshening in the upper ocean near Antarctica and increasing salinity below 800 m. In *zec750*, the main change in the salinity section is a freshening between 40 and 60° S in the upper 1000 m. In *zec2000* the ongoing salinity changes are more uniform, with a general freshening of the upper ocean above a band of increasing salinity below 500 m near Antarctica and extending north of 50° S at depths between 1000 and 2000 m.

Changes in the distribution of salinity are somewhat slower to become evident in *1pctCO2*. For instance, in the time series for the positions shown in panels j, k and l of Fig. 6, the salinity differences between *zec750* and the control are minor even after 300 years. Trends in temperature at the same positions were more distinct from the control and showed growing differences after 300 years. The transient response of salinity at 25° S and 250 m in *1pctCO2* is an increase in salinity. However, in all ZEC branches salinity decreases, albeit with significant interannual variability, indicating a recovery in the atmospheric circulation and precipitation under zero-emission climates with decreasing atmospheric $CO_2$.

The distribution and responses of ocean biogeochemical tracers (for example oxygen, Fig. 7) are distinct from both heat and salinity, due to the different distributions of tracer sources and sinks, both at the ocean surface and in the interior. Hence, the mean fields of biogeochemical tracers are distinct from physical tracers and are impacted in different ways by the changes in ocean state and circulation. As the strength of deep Antarctic overturning weakens, there is a decrease in the supply of oxygen from surface waters into all depths of the interior of Southern Ocean. Local exceptions to the general decline in oxygen include water between 0 and 1000 m at 50–60° S in low ZEC branches where oxygen likely increases due to the greater influence from southern oxygen-rich surface water and less from oxygen-poor waters because of changes in circulation and global stratification (panels e and h of Fig. 7). There is also an increase in subsurface oxygen below equatorial regions, north of 10° and below 500 m, where productivity declines in warmer climates of both *zec750* and *zec2000*. Reduced productivity and export of organic material reduce the consumption of subsurface oxygen in these regions driving this oxygen increase.

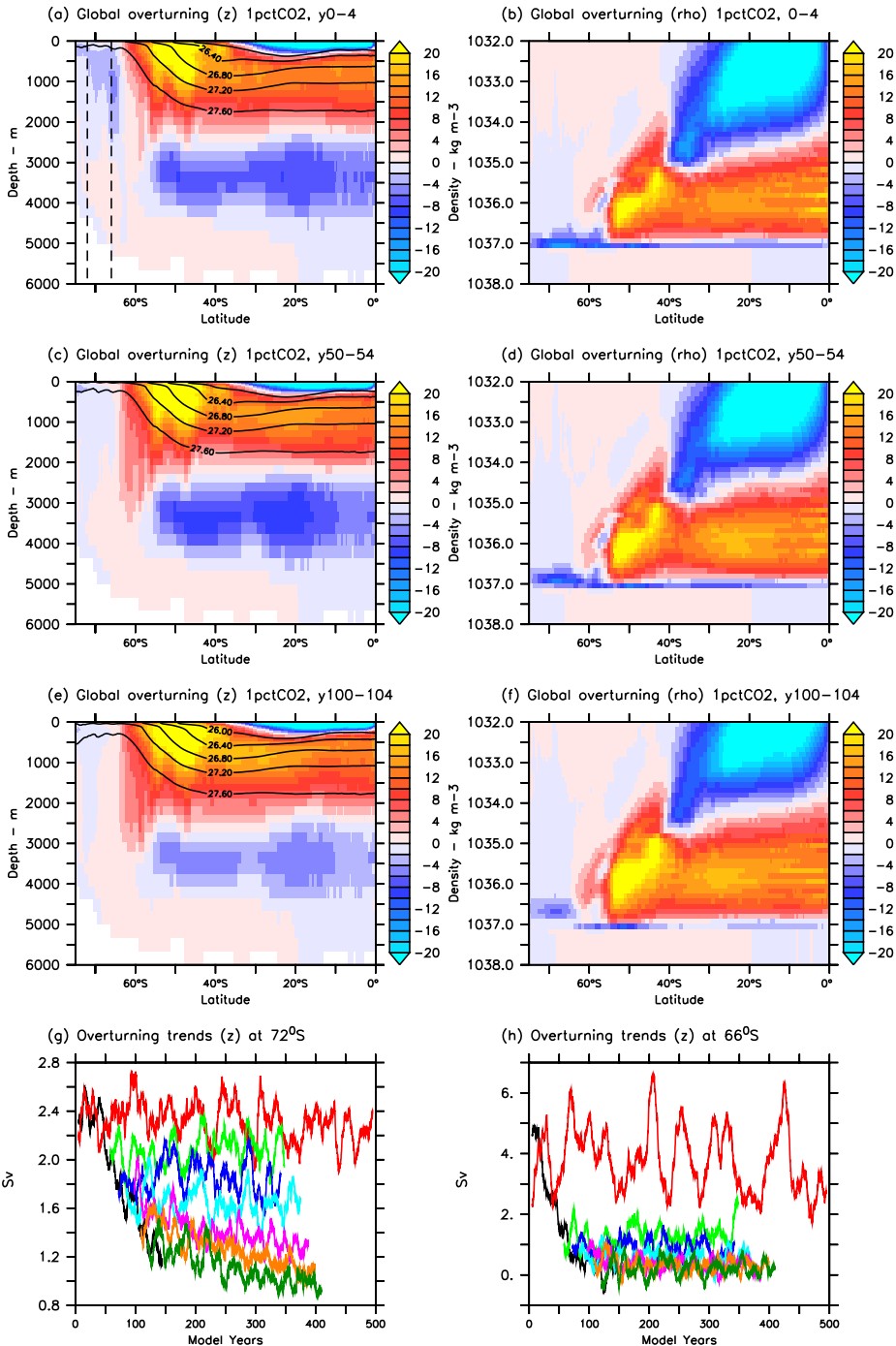

**Figure 4.** Global meridional overturning streamfunctions from *1pctCO2* calculated with respect to depth (left) and density (right, referenced to 2000 decibars). The streamfunctions shown are 5-year averages: first 5 years of *1pctCO2* (top row), years 50–54 (2nd row), and years 100–104 (3rd row). The bottom row shows time series of overturning calculated with respect to depth at two positions near Antarctica (at 72° S and 66° S, indicated by dashed lines in the top row), from *1pctCO2*, *piControl* and ZEC branches smoothed with a 10-year filter (using the same colour scheme as Fig. 1).

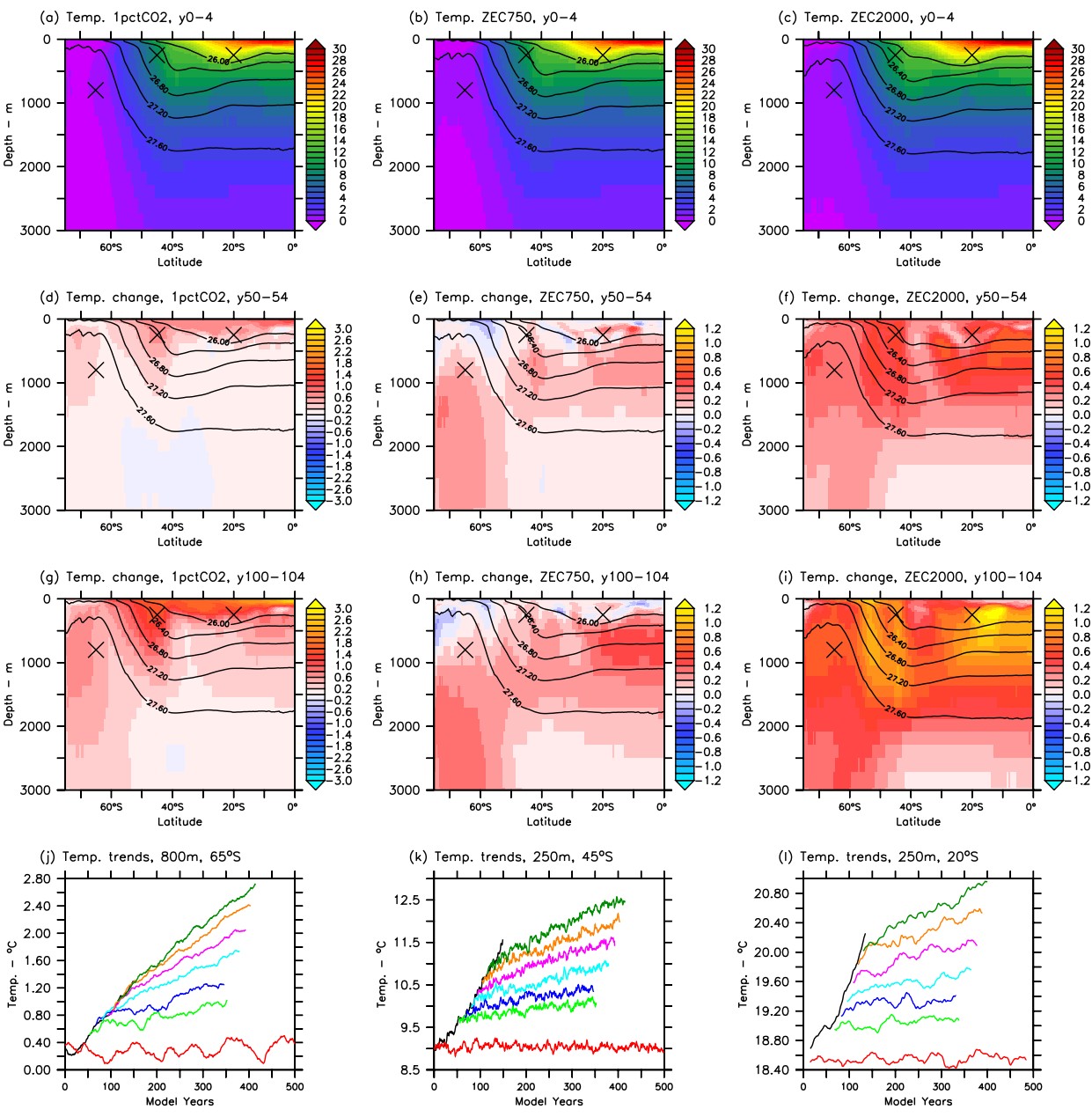

**Figure 5.** Changes and time series in average zonal sections of temperature. The top row shows zonally averaged sections for the first 5 years from *1pctCO2* (left) and ZECMIP branches after emitting 750 PgC (middle) and 2000 PgC (right). The second and third rows show changes in zonally averaged sections from the same experiments after 50 and 100 years, respectively. Contours indicate zonally averaged potential densities. The bottom row shows time series of subsurface temperatures in the Southern Ocean (at 65° S, 45° S and 20° S, at positions indicated), from *1pctCO2*, *piControl* and ZEC branches (same colour scheme as Fig. 1). Time series at 65° S and 45° S are filtered by 1 year, series at 20° S are filtered by 30 years.

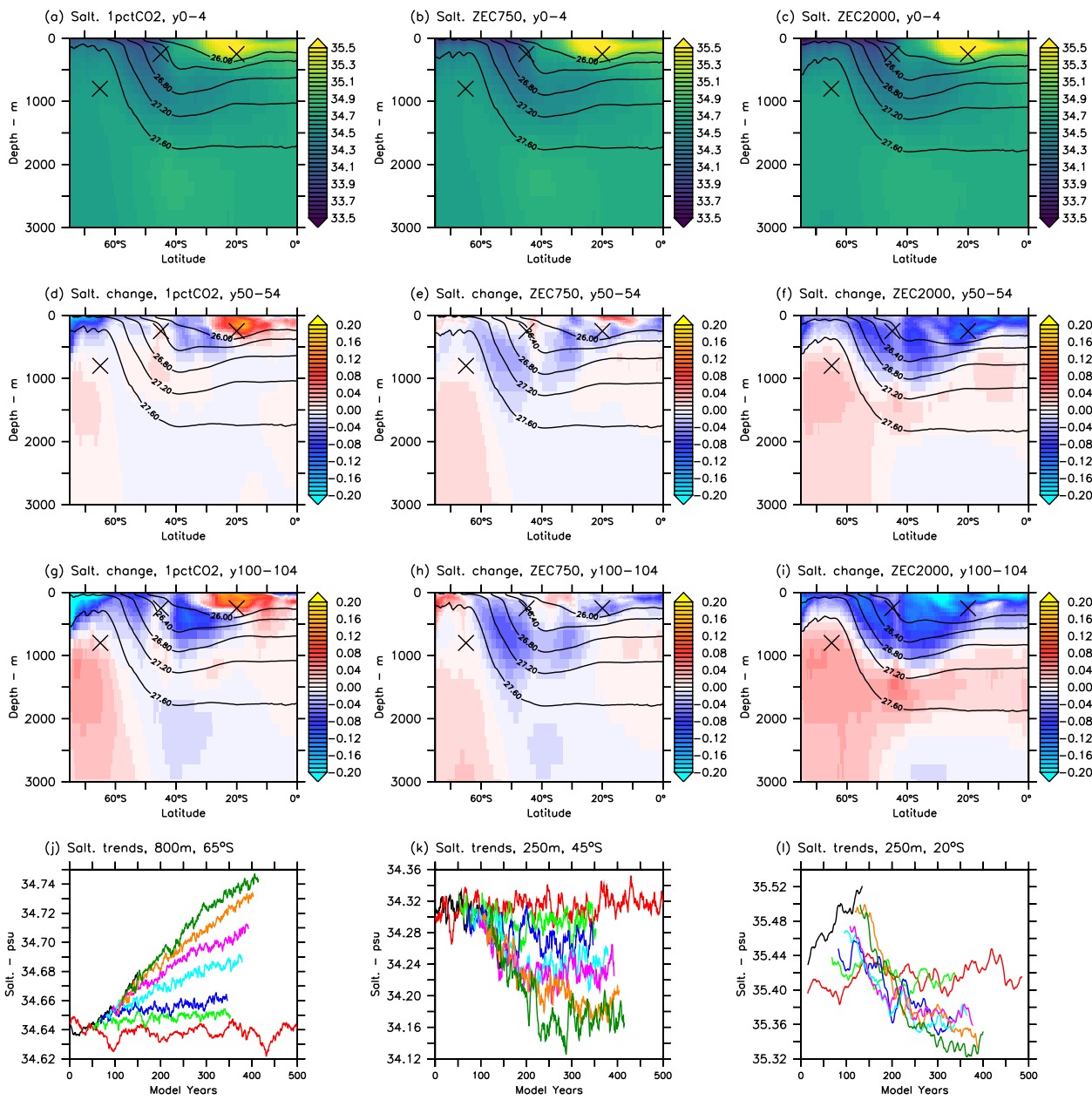

**Figure 6.** Changes and time series in average zonal sections of salinity, with the same layout as Fig. 5.

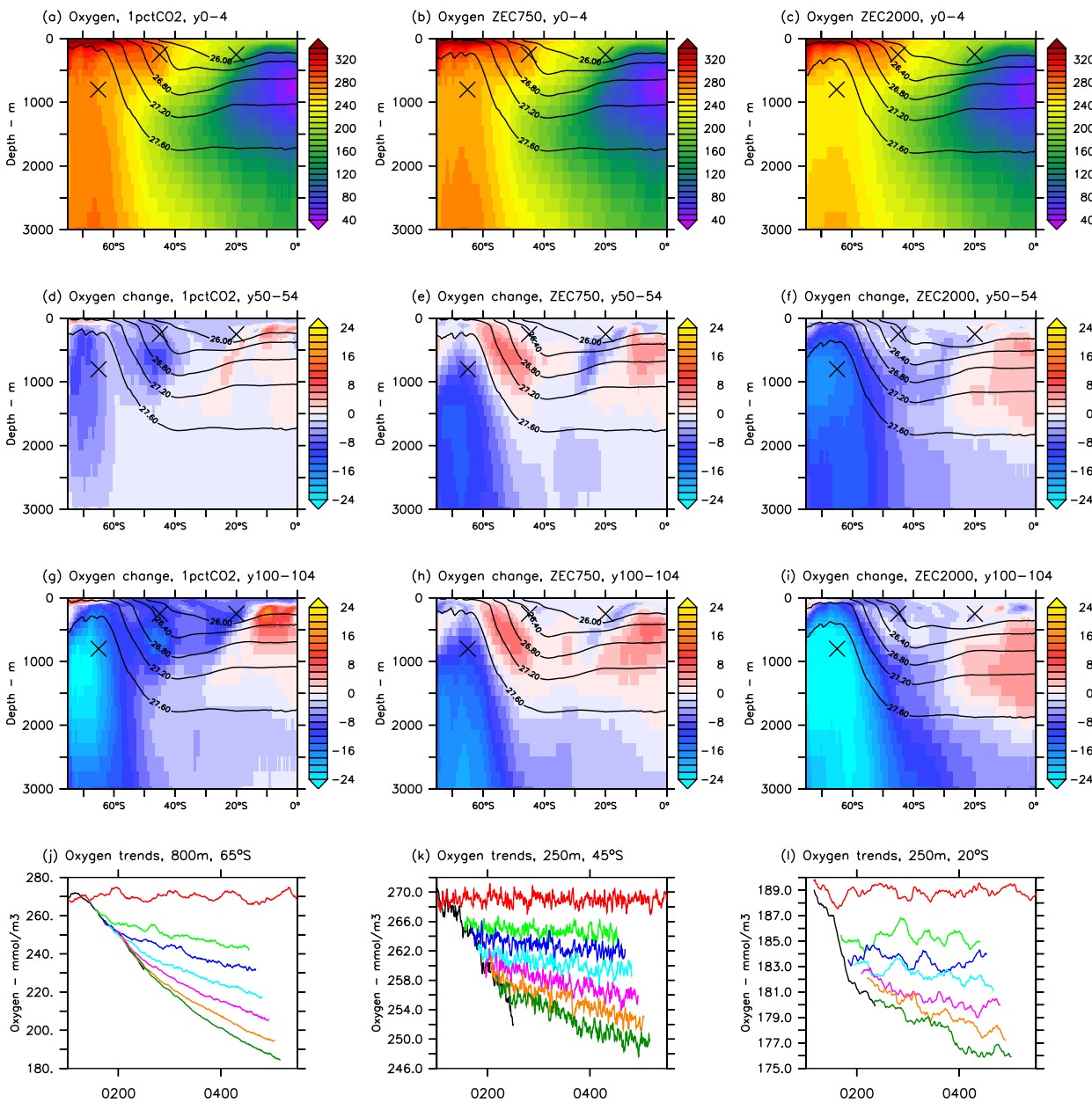

**Figure 7.** Changes and time series in average zonal sections of oxygen, with the same layout as Fig. 5.

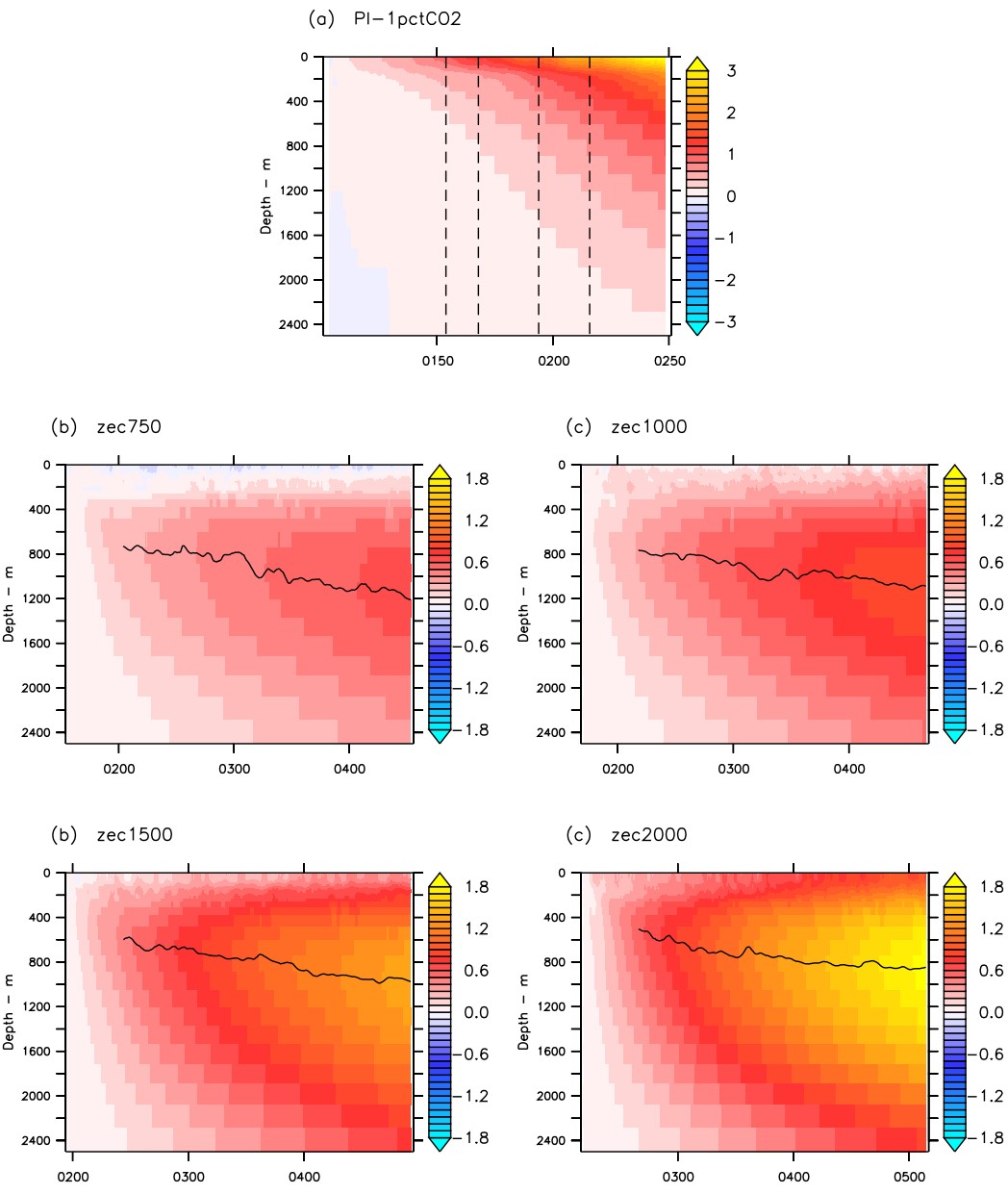

**Figure 8.** Heat uptake of the whole ocean as a function of depth and time, shown as changes in global averages of temperature within each experiment. Dashed vertical lines in panel (a) indicate the times that the ZEC experiments in (b)-(e) branch from the *1pctCO2*. Solid lines overlain indicate the depth of maximum change in temperature in ZEC branches.

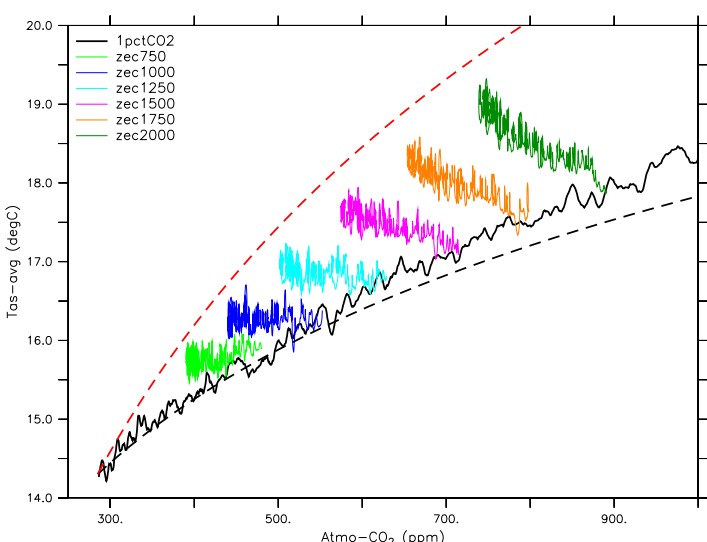

**Figure 9.** Time series of globally averaged surface temperatures with respect to atmospheric $CO_2$ for the *1pctCO2* and ZEC branches. Dashed lines indicate temperatures corresponding to the TCR (black) and ECS (red).

### 3.4 $CO_2$-temperature trajectories

Figure 9 shows the trajectories of *1pctCO2* and ZEC branches with respect to $CO_2$ and global average near-surface tempera­tures. The trajectories of these branches are consistent with climates approaching their equilibrium states after initial perturba­tions and warming in the *1pctCO2* experiment before branching. Overlying these experiments are temperatures corresponding to the Transient Climate Response (TCR) and Equilibrium Climate Sensitivity (ECS) of the ACCESS-ESM1.5, calculated with the logarithmic relationship of $CO_2$ and radiative or 'climate forcing' (Myhre et al., 1998) and assuming a constant climate

feedback parameter ($\lambda$, Wm$^{-2}$ °C$^{-1}$). Both TCR and ECS are expressed as the global warming associated with a doubling of atmospheric $CO_2$, and TCR is typically substantially less than a models ECS. The TCR and ECS for the ACCESS-ESM1.5 are 1.95°C and 3.87°C respectively (Ziehn et al., 2020). The trajectory of *1pctCO2* in Fig. 9 starts from the lower left (285 ppm, 14.3°C) and moves to the right, generally following the TCR (the TCR is defined by the response of *1pctCO2* at 70 years). As ZEC experiments branch their $CO_2$-temperature trajectories turn left with decreasing $CO_2$ and move towards the ECS over the

300 years of integration. Consistent with the time series of the surface temperatures in Fig. 1, the trajectories of the lowest ZEC branches have stabilised near their equilibrium climates by the end of the 300-year integration and are close to ECS values. Climate states of higher ZEC branches are still evolving and with further model integration are expected to also stabilise at an equilibrium climate temperature, though this may take several centuries, or longer for the highest branches.

## 3.5 Slab model

As a way to explain and understand the global temperature trajectories in *1pctCO2* and ZEC branches, a simple model of independant slabs with different inertias, conceptually representing responses from the land and ocean, is used to replicate these trajectories. There are other simplified models that have been constructed to emulate full ESMs, such as "energy balance models" (e.g., Geoffroy et al., 2013), though the slab model based on temporal responses is quite adequate to reproduce the average temperatures from the ACCESS-ESM1.5 here. Global temperature is an average of just two slabs that both approach the same equilibrium temperature anomaly determined by time-evolving atmospheric $CO_2$, as diagnosed from ACCESS-ESM1.5 simulations. Various processes related to the heat uptake and response of surface temperature for both the land and the ocean are parameterised in the timescales and inertias assumed. These global temperature trajectories are shown in Fig. 10, where the timescale of the land slab is 1 year and effectively follows the equilibrium temperature while the ocean timescale is 300 years. See App. A for more details and discussion on the model setup. These timescales for land and ocean were determined by fitting to global temperatures of *1pctCO2*. Temperatures of slab models with ocean timescales of 100 and 500 years are also shown which over- and underestimate the *1pctCO2* temperature time series. The slab model captures both the *1pctCO2* and the key trends of the ZECMIP trajectories shown, namely the slight decrease in *zec750*, neutral *zec1000* and increases in higher ZEC branches.

Being able to replicate the global temperatures with this slab model demonstrates that the ZEC trends found with ACCESS-ESM1.5 are due to the inertial response of the ocean within the climate system, which (from the zonal temperatures of Fig. 2) can be attributed to the Southern Ocean. In this way the Southern Ocean is like the "freight train" of the climate system; once the Southern Ocean has started warming noticeably in transient scenarios it will continue warming and even affect the global climate after switching to zero emissions (as shown in Fig. 2). At this point, the long-term global temperature trajectory will not be reversed by zero-emission scenarios but rather require ongoing negative emissions and extracting $CO_2$ from the climate system.

Climate "tipping points" can be considered as thresholds at which the climate changes to a new state and potentially continues to evolve without applying further increases in climate forcing; for example the loss of ice sheets or permafrost, forest die-back or shutdown of overturning circulations. Some of this tipping point behaviour is present in results presented here, though there are no new processes and the fanning out of global temperature trends in Fig. 1 indicate this does not occur at a single point as such, but rather it is a transition, where the later the branching off from the *1pctCO2* experiment, or more $CO_2$ emitted before switching to zero emissions, the stronger the ongoing warming. This general result does not preclude other local tipping points to be crossed in the process, notably changes to circulation and structure of the Southern Ocean during the warm epoch; for example, the point at which the average Antarctic sea ice area starts to decrease in Fig. 1f, or changes in overturning streamfunction in Fig. 4h. While the Southern Ocean and its climate response may not fit an example of a tipping point, its potential to drive ongoing warming with global impacts, supported by results of the slab model and without additional climate forcing, suggests it should be considered in discussions of regions and processes of particular interest for climate change.

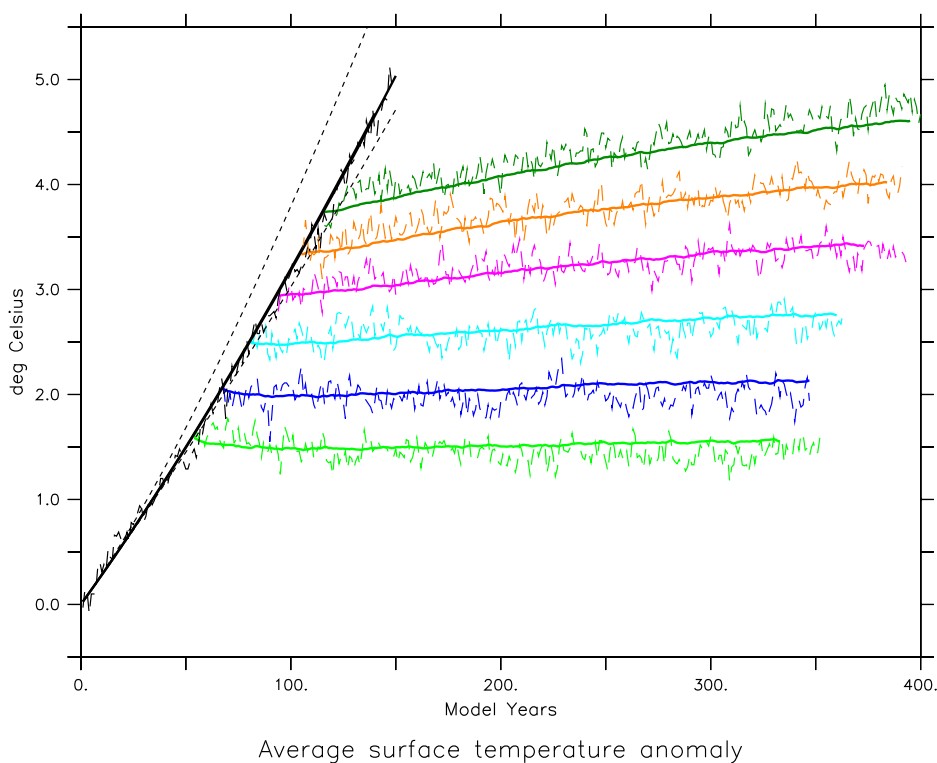

**Figure 10.** Global time series of average surface temperature for the *1pctCO2* and ZEC branches from ACCESS-ESM1.5 (dashed lines) and slab model (solid lines, same colour scheme as Fig. 9). Dotted black lines show slab model temperatures for *1pctCO2* with ocean time scales of 100 and 500 years.

## 3.6 Multi-model Comparison

### 3.6.1 Global surface temperatures

A curious observation from ZECMIP was the range of responses from the different models, such as in Fig. 6 of MacDougall
et al. (2020). In particular, how some models (ACCESS-ESM1.5 and UKESM1-0-LL) showed positive ZEC values in *zec2000* while other models (GFDL-ESM2M, MIROC-ES2L and CanESM5) were negative.

The slab model that is primarily tuned to ACCESS-ESM1.5 is now driven with $CO_2$ diagnosed from other ZECMIP ESMs that simulated all three ZECMIP type-A experiments to assess how the physical component of the ACCESS-ESM1.5 would respond if it were coupled with different biogeochemistry, and how much of the ZEC response is dependant on

the physical versus the biogeochemical components. Atmospheric $CO_2$ from each ZECMIP model, that are available at http://terra.seos.uvic.ca/ZEC (Eby, 2023), are used to force the slab model and slab-model temperatures are compared to the temperatures from the original ZECMIP models. Figure 11 shows results of these comparisons for the three ESMs that submitted output for all three ZECMIP branches. The ECS of the slab model for these comparisons is made to match the ECS of each model as reported in MacDougall et al. (2020); with the one exception for the GFDL-ESM2M where a higher value

of $2.9°C$ is used to approximately match the original *zec2000* temperatures instead of the reported $2.4°C$. This is consistent with Paynter et al. (2018) who found the GFDL-ESM2M had a higher ECS in multi-millenial simulations due to changes in the climate feedback parameter associated with ongoing evolution in sea surface temperature and atmospheric state. The slab ECS is adjusted primarily so results are on the same scale with the other models. Otherwise, the slab is tuned to the physical response of ACCESS-ESM1.5.

The first observation from Fig. 11 is the overall similarity of the response of the slab model with the temperatures found with original GFDL-ESM2M and UKESM results. In both of these models, the global temperatures in *zec2000* continue to rise on a centennial timescale, like the ACCESS-ESM1.5, despite the lower $CO_2$ values (Fig. 11a). While the centennial responses of the ZECMIP models are similar, their $ZEC_{50}$ values are quite different and even of opposite sign, as shown in Fig. 6 of the ZECMIP paper (MacDougall et al., 2020), and calculated again here (Table 2) and discussed below. These disparate results

can be associated with the models having different responses at shorter, annual to decadal timescales.

    The GFDL-ESM2M has the largest drawdown of $CO_2$ of the models shown (Fig. 11a) and there is cooling in both the original GFDL and slab models in the first decades of all ZEC branches (Fig. 11b). However, beyond 100 years, the centennial responses of the models dominate and temperatures rise in original GFDL-ESM2M results and the slab for *zec2000*. There is a similarity in the physical response of these two models in that there are similar relative trends in ZEC values for all branches

in Table 2. The long term ZEC of the GFDL-ESM2M is positive and increases like the original ACCESS-ESM1.5, despite different $CO_2$ responses. Note, both GFDL-ESM2M and ACCESS-ESM1.5 have MOM5 as their ocean component, albeit with different grids and parameterisations.

    The UKESM has a relatively slow temperature response to the changing $CO_2$ even in the course of *1pctCO2* where the original UKESM temperature increases are apparently delayed with respect to the slab (Fig. 11d). This lagged response is also

seen at the start of each ZEC branch, where temperatures continue increasing for the first decade, associated with the previous increasing $CO_2$ from before the branch points. Consequently, the $ZEC_{50}$ calculated with UKESM is significantly positive, even for the lowest *zec750* branch which otherwise shows a relatively neutral response on the centennial timescales in both the original UKESM and slab results. Slab ZEC temperatures with the UKESM $CO_2$ that do not have this lagged response are negative for *zec750* and *zec1000*, and positive to *zec2000* whereas UKESM ZEC values are all positive (Table 2).

The MIROC results are distinct relative to the other models here. The global temperatures in original MIROC results are decreasing on centennial timescales for all ZEC branches. The MIROC temperature response closely follows changes in $CO_2$ and climate forcing. In contrast, the slab tuned to the ACCESS-ESM1.5 has a slower response and shows rising temperatures in *zec2000* with the same MIROC $CO_2$. In Table 2, the $ZEC_{50}$ values of the ACCESS slab are similar to the original MIROC values, within $\sim 0.1°C$, whereas MIROC $ZEC_{200}$ values are 0.2 to $0.3°C$ lower than the ACCESS slab values.

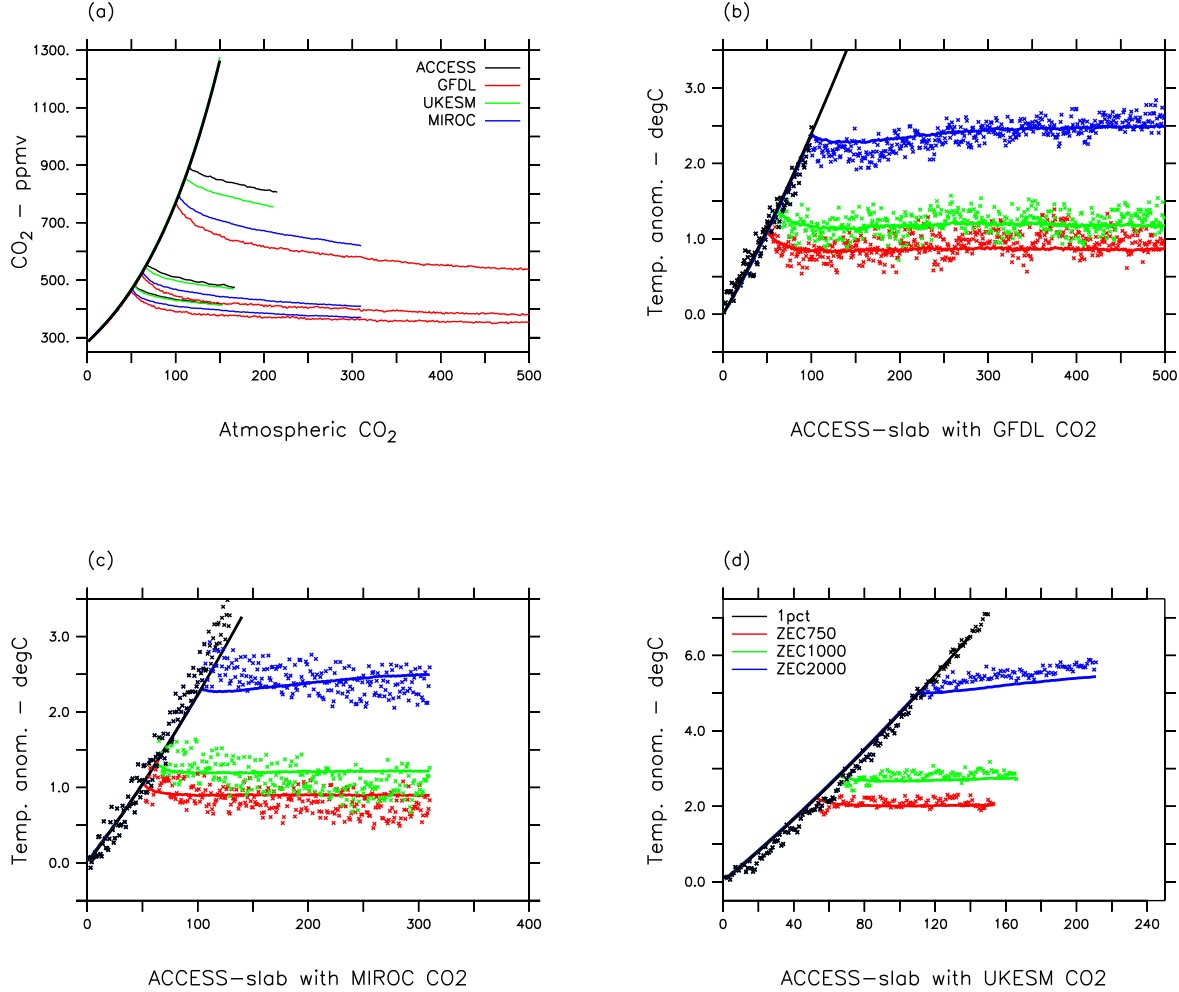

**Figure 11.** Global time series of (a) atmospheric $CO_2$ and average surface temperatures for the *1pctCO2* and ZEC branches from ESM models submitted to the ZECMIP: (b) GFDL, (c) MIROC and (d) UKESM. Annual averages of original model temperatures are shown as individual points; slab model temperatures are solid lines.

From these comparisons, the different long-term responses shown in ZECMIP models largely depend of properties of the physical climate models with some influence of the carbon cycle on the decadal responses and low ZEC branches; noting that while ZEC values from *zec1000* with ACCESS-ESM1.5 are zero (within uncertainty, Table 1), values from the slab tuned to ACCESS-ESM1.5 but with $CO_2$ diagnosed from *zec1000* experiments with other ZECMIP models are all negative (Table 2). Overall, ACCESS-ESM1.5, GFDL-ESM2M and UKESM are similar, showing significant responses at centennial timescales,

**Table 2.** ZEC values from ZECMIP model temperatures submitted to MacDougall et al. (2020), as well as ZEC values from the ACCESS-ESM1.5 slab with $CO_2$ diagnosed from ZECMIP models. Values are the differences between 20-year averages centred at the year of the ZEC branch (or 10-year average in the case of UKESM $ZEC_{100}$ values), relative to the 20-year average from the respective *1pctCO2* centred at the branch point.

| | | $ZEC_{50}$ | | $ZEC_{100}$ | | $ZEC_{200}$ | |
| --- | --- | --- | --- | --- | --- | --- | --- |
| | | Original | Slab | Original | Slab | Original | Slab |
| GFDL | *zec750* | -0.33 | -0.26 | -0.25 | -0.26 | -0.29 | -0.24 |
| | *zec1000* | -0.29 | -0.25 | -0.13 | -0.25 | -0.02 | -0.20 |
| | *zec2000* | -0.11 | -0.06 | +0.02 | -0.01 | +0.22 | +0.11 |
| MIROC | *zec750* | -0.17 | -0.19 | -0.24 | -0.18 | -0.36 | -0.19 |
| | *zec1000* | -0.05 | -0.16 | -0.23 | -0.15 | -0.36 | -0.14 |
| | *zec2000* | -0.08 | -0.03 | -0.13 | +0.05 | -0.23 | +0.15 |
| UKESM | *zec750* | +0.11 | -0.29 | +0.03 | -0.27 | n/a | n/a |
| | *zec1000* | +0.28 | -0.21 | +0.26 | -0.16 | n/a | n/a |
| | *zec2000* | +0.53 | +0.12 | +0.78 | +0.33 | n/a | n/a |

in contrast to MIROC where temperatures cool as atmospheric $CO_2$ decreases. Details in the physical response, such as the relative contributions at annual to decadal timescales, affect the values calculated for $ZEC_{50}$. The rapid $CO_2$ uptake of the GFDL-ESM2M lead to negative $ZEC_{50}$ values in each branch, in both the original model and the slab, whereas the lagged, decadal response in the UKESM produced positive $ZEC_{50}$ values in the original model but not the ACCESS slab.

These observations indicate that while $ZEC_{50}$ values are relevant on policy timescales, where modifications to current rates of $CO_2$ emissions may modify the expected $ZEC_{50}$, this metric can be a poor representation of the complete response of ESMs and later ZEC values are useful to consider for long-term implications to the climate state. Frölicher et al. (2014) also found variable responses over long integrations of ESMs under zero-emission scenerios, finding changes in the influence of the ocean on the global climate and even varying ECS values on multi-century timescales.

### 3.6.2 Zonal surface temperatures

In the case of the ACCESS-ESM1.5, the temporal evolution of zonal average temperatures, as shown in Fig. 2, clearly indicate the latitudes of the Southern Ocean as regions of a slow response in *1pctCO2* and also ongoing warming in all ZEC branches tested. Figure 12 is an equivalent plot of zonal average temperatures with available ZECMIP ESMs, namely MIROC-ES2L, the UKESM1-0-LL and ACCESS-ESM1.5. Note, these results are based on the experiments originally submitted to ZECMIP and available on the ESGF (World Climate Research Program, 2023). In the transient *1pctCO2* phase (left column), the broad patterns in the temperature changes are similar, each model shows greatest warming in the Arctic and slowest warming around the Southern Ocean, though Arctic warming is greater in the UKESM by several degrees. There is an overall global neutral

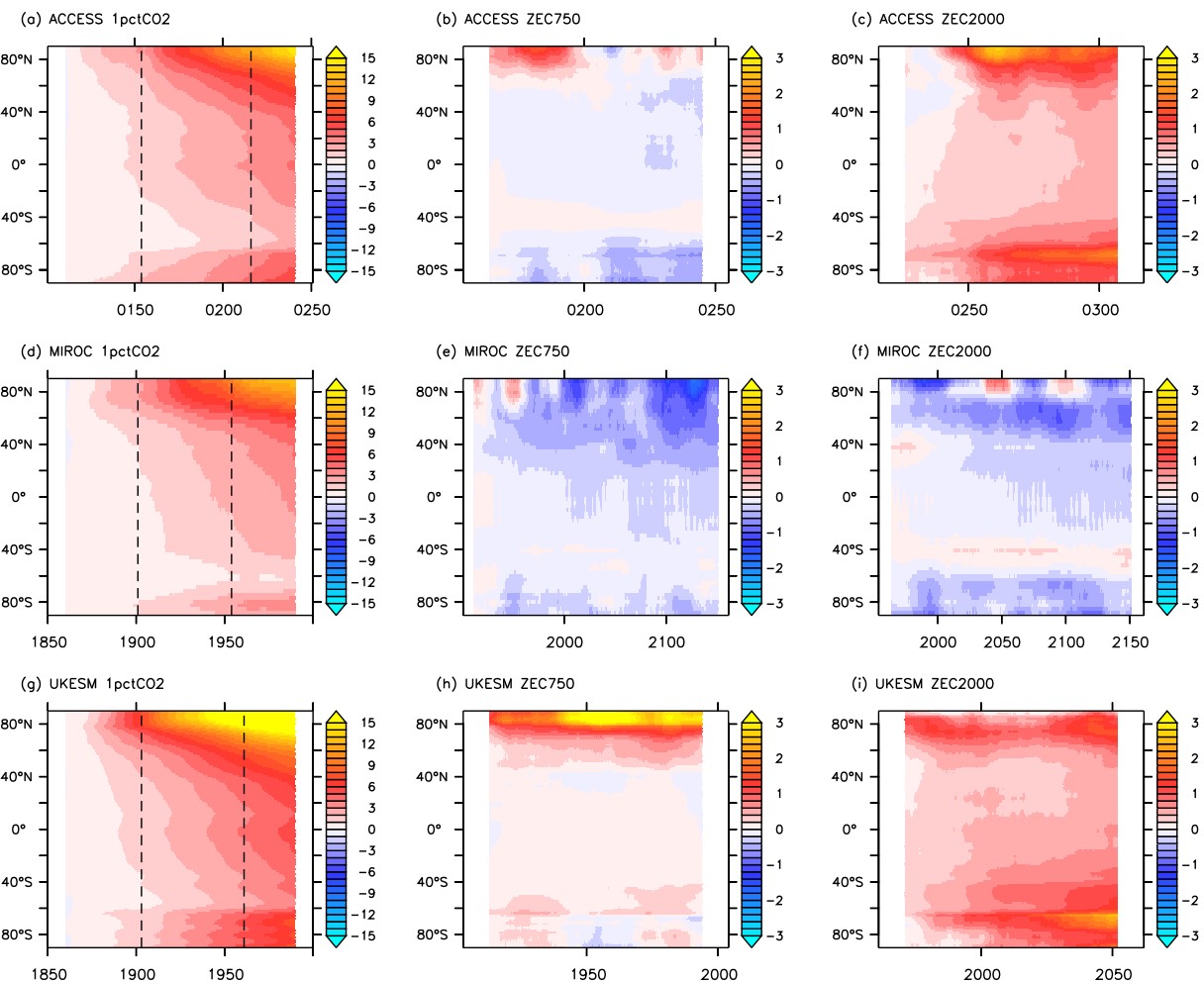

**Figure 12.** Changes in zonal average temperatures in *1pctCO2* (left), *zec750* (middle) and *zec2000* (right) from ZECMIP ESM models: ACCESS-ESM1.5 (top), MIROC (centre) and UKESM (bottom). Temperatures are smoothed by averaging over 20 years, and are differenced with respect to 20 years of *1pctCO2* centred at the branch point of each experiment, indicated by dashed lines of the *1pctCO2* panel for each model.

response in the 100 years of *zec750* for ACCESS-ESM1.5 and UKESM, and cooling in MIROC (middle column), and temperature changes related to slow modes of climate variability are evident. ACCESS-ESM1.5 showed warming at Southern Ocean latitudes in *zec750* before in Fig. 2, but this manifests on timescales longer than the 100 years shown here. In *zec750*, the Arctic region cools in MIROC but continues to warm in UKESM. In *zec2000* (right column), consistent with Fig. 11, there is broad warming in ACCESS-ESM1.5 and UKESM, but cooling in MIROC. The Southern Ocean features prominently as a region of ongoing warming in both ACCESS-ESM1.5 and UKESM, particularly at latitudes under the influence of sea ice, south of 60°

S. The Arctic in the UKESM shows less warming than the ACCESS-ESM1.5 in *zec2000*, though the UKESM has warmed more here in the transient experiment before branching. Even in MIROC, which shows overall cooling in *zec2000*, the Southern Ocean is a site of local warming, in this case at latitudes predominantly outside seasonal sea ice, 40–60° S; demonstrating that the response of the Southern Ocean to continue warming in high ZEC branches is common in all full ESMs.

## 4 Conclusions

The ACCESS-ESM1.5 submission to the recent ZECMIP (MacDougall et al., 2020) was one of two full-ESMs to test the zero-emission scenario after emitting 2000 PgC (*zec2000*) and demonstrate a significant positive ZEC value, or ongoing warming; another three ESMs simulated negative ZEC values and cooling. In contrast, ZEC has been assumed to be approximately zero for the present day climate state; for reference, the estimated total emissions of carbon between 1850 and 2022 is estimated to be 695±70 PgC (Friedlingstein et al., 2023), not accounting for affects of other climate forcing agents. Extra experiments with ACCESS-ESM1.5 have been executed to better understand the processes behind this ongoing warming, with more branch points after the emission of intermediate carbon budgets and also longer climate integrations out to 300 years with zero emissions.

The rates of ongoing global temperature increases vary smoothly across the ZEC branches, the increase is greatest on branches after the emission of the most carbon, and global temperature decreases slightly for the lowest branch, *zec750*. Longer integrations demonstrate significant regional changes that were not apparent in the original ZECMIP integrations. For instance, even in *zec750* there is a decline in Antarctic sea ice that is apparent after ∼200 years. Zonal time series of surface temperatures show that while the Southern Ocean is slow to warm in the transient *1pctCO2* experiment, this is the region that continues to warm in all ZEC scenarios, even in low ZEC branches and regardless of the global response.

Clear and persistent changes are evident in the subsurface ocean that start in *1pctCO2* and do not recover in any of the ZEC branches. The decrease in the Southern Ocean overturning circulation is associated with a decrease in density of the southernmost waters. These circulation changes then contribute to ongoing changes in the distribution of ocean tracers, both physical and biogeochemical. Heat increases at depth, even in low branches where there is cooling in surface waters. Biogeochemical responses are affected by changing circulation and changing source/sink terms. Oxygen decreases in the deep Southern Ocean in all branches with the decrease in overturning, but also increases locally at positions where reduced ocean productivity reduces consumption of subsurface oxygen. We note that some other models and studies that have assessed the climate in ZEC scenarios have identified significant responses in the North Atlantic associated with AMOC changes, for example, Schwinger et al. (2022) examined various scenarios with the Norwegian ESM. Also, MacDougall et al. (2022) investigated regional responses and found the most significant changes were also in the North Atlantic in some ZECMIP models, though the AMOC response was relatively weak in the ACCESS-ESM1.5. These ZECMIP-AMOC responses are not inconsistent with the Southern Ocean response presented here that is generally not prominent in the first century of low-ZEC branches.

The evolution of ZEC branches with the ACCESS-ESM1.5 with respect to atmospheric $CO_2$ and average surface temperature all traverse the space between the Transient Climate Response (as followed by the *1pctCO2*) and the Equilibrium Climate

Sensitivity. In this space, the ECS is significantly higher than the TCR, so it is not unreasonable for a climate to be warming even with decreasing $CO_2$ while the climate states traverses from a transient response towards its equilibrium state.

Global trajectories found with the full ACCESS-ESM1.5 are well reproduced with a simple composite slab model, where each slab approaches the same equilibrium temperature change prescribed by the climate forcing with a different timescale. The ongoing temperature increases are explained by the slow response of the ocean. The Southern Ocean in particular behaves as the "freight train" of the climate system; once the Southern Ocean starts warming significantly it will take a large change in the climate forcing, such as a substantial reduction in $CO_2$ beyond the natural uptake of land and ocean, in order to reverse its temperature trajectory and its affect on the global climate.

This slab tuned to the ACCESS-ESM1.5 is then forced with $CO_2$ diagnosed from other ZECMIP models to evaluate whether the positive ZEC from the original ACCESS-ESM1.5 *zec2000* experiment is due to the physical or biogeochemical component of the model. The stronger $CO_2$ drawdowns from these other models, relative to ACCESS-ESM1.5, reduce the ZEC values calculated in Table 2. Most $ZEC_{50}$ values are now negative with the ACCESS-ESM1.5-tuned slab. However, the centennial response with these *zec2000* $CO_2$ pathways are still similar with positive and increasing slab temperatures.

This slow, centennial response of the Southern Ocean in ZEC scenarios is a common feature of climate models, and it has been identified in the observations of the ocean over recent decades (Armour et al., 2016). It is present in the zonal surface temperature of other ZECMIP models shown here in Fig. 12; even MIROC which is cooling globally is warming at Southern Ocean latitudes in *zec2000* (Fig. 12f). Gillett et al. (2011) demonstrated the Southern Ocean continued to warm in their 1000-year simulations with the Canadian ESM. Supplementary figures of Schwinger et al. (2022), their Fig. S2a, also showed warming at Southern Ocean latitudes south of 40°S after 200 years in all the ZEC and overshoot scenarios tested. It appears that the centennial response of the Southern Ocean and ongoing warming under zero emissions is a common feature of climate models and would be expected in the real Earth climate system. The magnitude of this response does vary and in some models, including the ACCESS-ESM1.5, it is having a significant global impact.

*Data availability.*

CMIP6 output from the ACCESS-ESM1.5 experiments, and from other models that submitted results to the original ZECMIP analysis, is freely available through the Earth System Grid Federation (World Climate Research Program, 2023), including the *piControl*, the *1pctCO2* and the original branch experiments submitted to ZECMIP: *zec750*, *zec1000* and *zec2000*. $CO_2$ values from ZECMIP models used to drive the ACCESS slab model were obtained from the ZECMIP data repository (Eby, 2023). For output related to the extra experiments described in the manuscript, please contact the authors.

**Appendix A: Slab model**

In Section 3.5, time series in global temperature are compared with a simple model (Fig. 10) composed of slabs with different "thermal inertia," or slabs that respond to changes in climate forcing on different timescales. This slab model is also driven with

**Table A1.** Components of the slab model in presented in Fig. 10.

| Slab | Fraction | ECS $^\circ$C | Timescale years |
|---|---|---|---|
| Land | 0.5 | 3.87 | 1 |
| Ocean | 0.5 | 3.87 | 300 |

results from other ZECMIP models (Fig. 11). In this slab model, the temperature of each independant slab ($T_i$) tends towards the equilibrium temperature ($T_{eq}$) that is a function of atmospheric $CO_2$ with a prescribed timescale ($\tau_i$),

$$\frac{dT_i}{dt} = \left(T_{eq}(CO_2) - T_i\right)/\tau_i \tag{A1}$$

The global temperature is then a weighted average of the slabs ($T_{av} = (\sum_i w_i T_t)/\sum_i w_i$). These temperatures are anomalies with respect to preindustrial conditions.

     The equilibrium temperature is determined by the Equilibrium Climate Sensitivity ($T_{ECS}$, the change in equilibrium temperature with a doubling of atmospheric $CO_2$ from preindustrial, $CO_2^{PI}$, diagnosed with the method described in Gregory et al., 2004) and the atmospheric $CO_2$ diagnosed from ACCESS-ESM1.5 experiments or other ZECMIP ESMs. All other climate
forcing terms (e.g. aerosols and non-$CO_2$ greenhouse gases) in these experiments are held constant at preindustrial values. Climate forcing, or radiative forcing perturbations in Wm$^{-2}$, is proportional to the logarithm of atmospheric $CO_2$ (Myhre et al., 1998), so the equilibrium temperature can be determined from

$$T_{eq}(CO_2) = T_{ECS} \frac{\ln(CO_2/CO_2^{PI})}{\ln(2)} \tag{A2}$$

     Table A1 describes the slabs used here to replicate the ACCESS-ESM1.5 global temperature time series in Fig. 10. The
idealised "slab" model is intentionally kept simple while replicating the global trends from ACCESS-ESM1.5, and here two slabs meet this objective, conceptually corresponding to the response the land and ocean. The timescale of the "land"' response ($\tau$=1) effectively means the land follows the equilibrium temperature here. Note, the land weighting of 0.5 is significantly higher than the areal fraction of land over the real Earth. However, there is no intent to interpret these slabs to represent actual land temperatures, rather, their influence on the global temperature. Also, for the "ocean" slab only a single $\tau$ is applied when
in reality different regions of the ocean will respond differently to changes in climate forcing (such as the well-mixed Southern Ocean relative to the stratified tropics), and the single value represents a blended response of these varying oceanic components balanced with the terrestrial response.

     Other processes could be considered in the construction of this slab model, such as heat exchange between the slabs and/or the additional of extra slabs (a slab with a decadal timescale for example). However, given that the two-slab model effectively
reproduces the temperature time series in Fig. 10 these options are not necessary for the purposes used here.

To produce the trends of ZEC branches shown in Fig. 10 and 11, each $T_i$ starts from a temperature anomaly of 0°C, or the preindustrial state, and evolves along the trajectory defined by the $CO_2$ from the *1pctCO2* to the branch point, where it then tends towards the temperatures determined by the atmospheric $CO_2$ diagnosed from ACCESS-ESM1.5 or ZECMIP model experiments for each ZEC branch.

*Author contributions.*

TZ, MC and RL contributed to the development of the model. TZ ran experiments. MC analysed output and prepared the manuscript, all authors provided comments.

*Competing interests.*

The authors declare that they have no conflict of interest.

*Acknowledgements.* This research used computation resources and archives available at the National Computational Infrastructure (NCI), which is located at the Australian National University and supported by the Australian Government.

MC, TZ and RL receive funding from the Australian Government under the National Environmental Science Program (NESP).

The authors thank Andrew MacDougall and an anonymous reviewer for comments that have improved the paper.

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
