# Peer review of "The Southern Ocean as the climate's freight train – driving ongoing global warming under zero-emission scenarios with ACCESS-ESM1.5"

_Biogeosciences, 2023_

## Author Comment (AC1)

**Comment on bg-2023-146**

Andrew MacDougall

Review of: The Southern Ocean as the freight train of the global climate under zero-emission scenarios with ACCESS-ESM1.5

Overall assessment:

The paper documents experiments with ACCESS-ESM1.5 that expand upon the standard ZECMIP A-class experiments to better show the transition from negative ZEC at low cumulative emissions to positive ZEC at high cumulative emissions. Additionally that paper uses analysis of ACCESS-ESM1.5 and a slab ocean model to show that in ACCESS-ESM1.5 ZEC is dominated by Southern Ocean processes. While the paper is interesting, generally scientifically sound, and well written - some revisions are needed before publication.

Thank you for your review and comments.
These have been useful to prepare to improve upon the submitted manuscript and clarify the presentation and discussion of the results.
Please see below for details (in blue) regarding how we can address these points.

General Comments:

(1)  The authors appear to be using two different algorithms to compute ZEC values. In the caption for Table 2 the authors indicate that they are using the standard algorithm outlined in MacDougall et al. 2020. That is "Values are the differences between 20-year averages centred at the year of the ZEC branch [...], relative to the 20-year average from the respective 1pctCO2 centred at the branch point." While the caption for Figure 2 says the regional ZEC is being computed as "Differences are with respect to the average of the first 10 years of each experiment, and smoothed with a 5-year filter."

I strongly recommend that the authors compute regional ZEC using the same algorithm as global ZEC, using maps of the 20-years average temperature from the 1pctCO2 experiment centred on the year emissions cease as the cessation temperature reference value. Using a different algorithm to compute regional ZEC risks making the results of this study incomparable with other similar studies.

These figures have been regenerated with 20-year averages, referenced to *1pctCO2*, as suggested, with no major changes in the results shown.
There are some minor changes to the regional/zonal temperature changes in the *ZEC750* panels of Fig. 2 and 3 in the Arctic where there was decadal variability seen in the original version of the figures.

Additionally, a description of the algorithm used to compute ZEC should be included in the methods section.

A description of the ZEC calculation is now included in a ZECMIP subsection of the Methods.

(2) Section 4.2 "Multi model Comparison" is the least convincing part of the study. From Figure 11 it is clear the using the slab model tuned to  ACCESS-ESM1.5 does not capture MIROC or UKESM temperature trajectories well. While the match to GFLD is better both ESMs use the same ocean model (MOM) so a better match is to be expected. Additionally recent analysis of regional ZEC (MacDougall et al. 2022) showed that for a least some ESMs AMOC is dominating the ZEC response not the Southern Ocean, with some models having regional ZEC dominated by AMOC collapse (CESM2). Thus I suggest the existing section 4.2 be deleted and a more qualitative comparison be made to the regional ZEC effects shown in MacDougall et al. 2022.

The purpose of the slab model and this sub section will be clarified in the manuscript.
By forcing the slab model with $CO_2$ from other models, we can pull apart the physical and biogeochemical responses of the climate systems under zero-emission trajectories.
No, the slab does not reproduce other models, it wasn't meant to as it is showing how the ACCESS-ESM would respond with the carbon-cycle response of the other models. That there is a difference between models demonstrates it is differences in the physical models largely determining the zero-emission response, not the carbon cycle.
This may have been poorly communicated in the submitted manuscript, which can be modified to clarify these points.
Discussion is also added of other papers, in particular of the papers MacDougall et al. 2022 and also Schwinger et al. 2022 which has been brought to our attention and also discusses potential impacts of a collapse in the AMOC.

Aside, if the long-term ZEC response of the ESM is primarily determined by the physical component of the model, perhaps there is value in having a prescribed *ZEC1000* $CO_2$ time series, for example, to enable non-ESM climate models to run pseudo-ZEC experiments, much like a ZEC-scenario experiment.
This may assist in evaluating the relative roles of the AMOC and Southern Ocean and potentially other processes.

(3) Throughout the manuscript 4 digit model codes for time are used instead of years. Model years in all figures, tables and in text, should be given in standard Arabic numerals (no leading zero) with appropriate units (years). For figure captions please include a x-axis label of 'Model Years', to be clear that Gregorian calendar years are not being used.

Suggestions have been adopted.
Actually, the model does run with a Gregorian calendar (an unnecessary detail here), but yes, "Model Years" is a better way to label these axes.

(4) Please use consistent notation for the 1% CO2 experiment.

This experiment was referred 3 times in the submitted abstract; this will become once or twice after the abstract is revised to focus more on the conclusions rather than technical details.
We use a longer experiment name for readability in the abstract.
In the main text, the experiment should be referred to with the (somewhat) shorter format, *1pctCO2*.

Specific Comments:

Line 1: "Climate Projection Experiment" is an odd start. "Climate model simulations" would be more consistent with past terminology.

We use terms "simulation" and "experiment" almost synonymously, with some preference for "experiment."
To me, "experiment" is more defined and "simulation" is somewhat generic, and for results presented here "experiment" feels better.

Line 5: Delete 'Multi'

Thanks, deleted

Line 16 to 20: Odd framing since ECS becomes a moving target as CO2 concentration will not stabilize for a very long time. Processes not included in ESMs such as the CO2 weathering feedback will cause a slow drawdown to close to pre-industrial over the next few 100,000 years (e.g. Archer 2005). Statement makes more sense after reading the paper but not best framing for an abstract.

At the time, ECS and TCR seemed to be a succinct way to argue that positive ZEC values are entirely reasonable.
However, agreed, this doesn't work as well when it requires substantial explanation and can be dropped as the abstract is reworked to better highlight the main conclusions of the work..

Line 24: Delete "safe".
Deleted

Line 26: Change "greenhouse gas" to 'forcing agent' (aerosols are not gases), and add 'unaccounted for' before 'climate feedbacks'
Manuscript modified as suggested.

Line 93: Seems to indicate the ACCESS-ESM1.5 does not have dynamic vegetation. Should say this in model description. Many of the models that participated in ZECMIP did have dynamic vegetation so could be re-written to say this is the implementation in ACCESS-ESM1.5 but other models have prognostically changing vegetation maps.

To simplify what is meant to be a brief description here of the set up of the *1pctCO2* experiment, "vegetation" has will be dropped and this statement will just refer to land use being set to preindustrial conditions.
Yes, there is no dynamic vegetation in ACCESS-ESM1.5, but ACCESS-ESM1.5 has prognostic leaf-area-index, (see the model description paper, Ziehn et al. 2020) so the influence of a vegetation type is able to increase or decrease, though these details are not necessary here.

Line 96: I never liked the description of this as 'unrealistic'. An asteroid strike, global thermonuclear war, or supervolcano eruption, would probably do the trick. Add 'baring global cataclysm' before 'a global instantaneous', and change 'unrealistic' too 'unlikely' to fix.

The aerosols kicked up by these catastrophes will also complicate climate forcing. Perhaps a zombie apocalypse would be a plausible way to trigger a cleaner zero-emission scenarios...

That said, the suggested text is adopted to clarify the applicability of the experiments presented.
The intention here is to highlight that while the experiments have an idealised nature, the results are consistent with other climate stabilisation experiments with varying combinations of climate forcing components, where the temperature of the branching is a key factor that determines the ongoing climate trajectory.
The language is softened and modified to point out that actually, the straight-forward nature of these zero-emission experiments is powerful in that it is easy to replicate with different models and compare with future generations of CMIP.

Line 109: Change 'about linear' to 'approximately linear'. Change 'gradient' to 'trend'. A gradient usually indicates a change in space, not time.
Modified to "The time series of surface air temperatures from each ZEC branch are approximately linear. The overall rates of change are…"

Line 110: Tipping points can be subtle transitions.
This phrase is removed from here.
There was a time when we first looked Antarctic sea ice extent of 100 years from the original *ZEC750*, *ZEC1000* and *ZEC2000* experiments, that there did seem to be step change that might have been a tipping point behind the global response.

But alas, the story got more complicated after running the infill experiments, running experiments for longer and the fact the global results can be reproduced with this slab indicate the no tipping point is required. However, none of this is necessary at this point in the manuscript and the term will be removed from the manuscript.

Line 117: change 'of average' to 'in average'
Modified as suggested.

Figure 1d: The preindustrial TOA energy balance seems to be negative. Is this plotted correctly? If so, is the model drifting?
You observe correctly, there is some energy leakage somewhere in the coupled model and has been discussed in the model description paper (Ziehn et al. 2020).
However, the physical climate configuration of the ESM has been stable for some time and there's a 1000+ model years of spinup so this imbalance doesn't drive significant drift in any of the climate components.
Text will be added to the manuscript to state there is no drift associated with the offset in Fig. 1d.

Line 141 to 149: Could compare results to MacDougall et al. 2022 here.
A paragraph making a comparison to the temperature change maps of MacDougall et al. 2022 will be added to the text here. This will include a mention that some of the models show AMOC responses and also point out that the Southern Ocean response we are presenting here is not apparent because they manifest after more than 100 years after branching and after the $ZEC_{50}$ maps of MacDougall et al. 2022.

Line 256: May want to add a few sentences to discuss the potential impact of Ice Sheet loss on the Southern Ocean Feedbacks. I don't think ACCESS-ESM1.5 has a dynamic Ice Sheet model, so discussing what impact Ice Sheet loss may have is important.

No ACCESS-ESM doesn't have an active an ice sheet.
Sentences will be added to state this and the potential impact to further reduce overturning, as demonstrated in some recent ocean model results (Li et al 2023, [https://doi.org/10.1038/s41586-023-05762-w](https://doi.org/10.1038/s41586-023-05762-w)) who tested the impact of freshwater fluxes from Antarctica in climate change scenarios which decreased overturning and warmed the subsurface.
These statements may be placed in the subsection presenting overturning streamfunctions.

Line 319: Spell out ESGF the first time you use it.
Done, though the first use will now occur earlier in the manuscript.

Line 341 to 342: Need to note that this is 2 of 4, not 2 of 9.
Text is modified as suggested: "…was one of only two full-ESMs that demonstrated significant positive ZEC values, or ongoing warming, out of the four ESMs that tested the zero-emission scenario after emitting 2000 Pg"

Line 343: Change 'about neutral' to 'approximately zero'
Changed.

References:

Archer D. Fate of fossil fuel CO2 in geologic time. Journal of geophysical research: Oceans. 2005 Sep;110(C9).

MacDougall AH, Frölicher TL, Jones CD, Rogelj J, Matthews HD, Zickfeld K, Arora VK, Barrett NJ, Brovkin V, Burger FA, Eby M. Is there warming in the pipeline? A multi-model analysis of the Zero Emissions Commitment from CO 2. Biogeosciences. 2020 Jun 15;17(11):2987-3016.

---

## Author Comment (AC2)

**Comment on bg-2023-146**

Anonymous Referee #2

Chamberlain et al. present a set of idealized Earth system modelling experiments exploring the response of the Earth system to phasing out CO2 emissions. The experiments follow the "Zero Emission Commitment MIP" protocol but additional simulations (additional levels of emissions) are provided. The authors construct a simple slab (restoring) model where temperature is restored towards the equilibrium temperature of the model with two different time scales. They calibrate this model to their ESM and also compare results from three other ESMs with this slab model. The main finding that the authors emphasize is that the Southern Ocean continues to warm on centennial time scales after emissions ceased.

The response of the Earth system to phasing out emissions and the committed climate change due to prior emissions is a highly relevant research area, and there is generally a lack of ESM simulations that explore such scenarios. The model experiments presented here are therefore highly relevant and the model simulations are interesting and well designed. However, the manuscript reads in large parts like a technical summary of simulation results rather than focusing on new insights. Also, it remains unclear what new insights the two-slab model brings, particularly when it comes to the multi model comparison (see below for more details). Finally, the authors make no attempt to place their study in the context of previous literature, neither in the introduction nor in the discussion/conclusions section. Given these concerns, I would suggest substantial revisions of the manuscript before it might be suitable for publication in Biogeosciences. I cannot address all these points in depth in this review, but I will give some suggestions below.

Thank you for your review, fresh perspective and comments.
These have been useful to prepare to improve upon the submitted manuscript and clarify the presentation and discussion of the results.
Please see below for details (in blue) regarding how we can address these points.

Major points:

1) The abstract and the introduction contain too many technical details. For example, the abstract describes the ZECMIP simulation design (emission levels and the fact that the zero emission simulations are branched from the 1pctCO2 simulation), but is missing a summary of main results. The introduction is missing an account of previous literature (see below). In the results section, zonal mean section of salinity and oxygen are presented, but these results are never used or discussed (there are a few general sentences on biogeochemical changes in the conclusions section). It remains unclear what insights we gain from these figures, and how this relates to the main topic (the committed warming) of the paper. In general, the manuscript does not have a clear direction (at least it was not obvious to me). What do the authors want to address? Is it the fact that the ZEC750 simulation cools while the ZEC2000 simulation warms in their model? Is it a comparison with other models? Why do some models warm while others cool? What is the role of the Southern Ocean in this? It would help to formulate one or two clear research questions, and really set out to answer these.

This work presented is motivated by our recent participation in the ZECMIP and to understand why the model we used, ACCESS-ESM1.5 exhibited ongoing warming in some of the branches which was initially counter intuitive, but further analysis, with the help of the extra experiments presented, are able to demonstrate that it is the Southern Ocean driving a long-term global trend, which has not been featured much in the literature (but yes, as pointed out, the Southern Ocean is discussed by Gillet et al. 2011).

Salinity and other biogeochemical tracers were originally assessed when initially studying these experiments, testing if there were any indicative changes associated with the change in the sign of the ZEC values between the experiments, particular for salinity as there is potential to feedback on the thermohaline circulation.

Now these other tracers are secondary to the main results presented in the submitted manuscript, but they have been retained in the manuscript as further examples of changes to the climate state under zero-emission scenario.

Our motivations will be clarified by rewriting throughout the manuscript, including the abstract and conclusions.

2) Related to point 1, it also remains unclear what insights we gain from the application of the 2-slab model. I can see that the model is able to reproduces the global average surface temperature of the ZEC branches of the ACCESS model (no warming for ZEC750, increasingly more committed warming for higher emissions), but what does this mean physically? When it comes to the multi-model inter-comparison, the slab-model (as the authors present it) is not able to reproduce the cooling characteristics of the MIROC model, and the fit is not very good for the GFDL and UKESM models either. So again, what can we learn from this model then? As a side note, I believe that if there is something to learn from the slab-model, the model "ocean" time-scale would need to be fitted to the individual models (the authors only adjusted ECS). Would this improve the slab-model results? Would the slab model be able to reproduce the cooling in MIROC? As the results stand now, the slab-model seems to be able to reproduce the ACCESS ZEC-simulations more or less by chance, and it fails to reproduce relevant aspects of the zero emission commitment for the other models.

The purpose of the slab model is two-fold. Firstly, by replicating the global ACCESS-ESM results we show that the change in the global response can be understood by slow response of the ocean, the Southern Ocean in particular, to the climate forcing and no new processes need to be invoked. Secondly, by applying the slab model to $CO_2$ diagnosed from other models, we can pull apart the physical and biogeochemical responses of the climate systems under zero-emission trajectories.

No, the slab does not reproduce other models, it wasn't meant to as it was only tuned to the ACCESS-ESM and the slab emulates how the ACCESS-ESM would respond with the carbon-cycle response of the other models (the adjusting of the ECS is only to put the slab results on the same scale as the other models, as will be noted in a revised manuscript). That there are significant differences between the slab and the other models demonstrates it is differences in the physical models determining the overall zero-emission response, not the carbon cycle.

This may have been poorly communicated in the submitted manuscript, which can be modified to clarify these points, in the discussion here but also the abstract and conclusions.

3) This study is not the first to investigate the response of the Earth system to phasing out emissions, but neither the introduction nor the discussion/conclusions sections place the present manuscript in the context of previous literature. The first (to my knowledge) ZEC study with a full ESM was by Gillett et al. (2011), who also emphasize changes in the Southern Ocean. The study by Frölicher et al. (2014) finds a pronounced multi-centennial warming in one model, while a second model shows a cooling trend. Recently, Schwinger et al. (2022) have conducted a study, which also was based on the ZECMIP protocol, and they find a dominant role of AMOC decline and recovery for ZEC in their model. These studies come to my mind immediately, but there are probably more.

The work presented here is motivated by participation in ZECMIP which in itself included contributions from multiple models.

Thank you for indicating the other papers, these will be added along with another recent ZECMIP analysis paper to further improve the manuscript. The results in these are broadly consistent with our own results and our comparisons with other ZECMIP models and the inter-model variability found.

Schwinger et al 2022 in particular will be an interesting comparison which like experiments presented here, also explore a range of zero-emission branches (and overshoot scenarios) up to the emission of 2500 PgC

and test the climate response (and recovery) over multiple centuries, using the Norwegian ESM and a physical climate configuration that is sensitive to the AMOC (their supplementary Fig. S2 shows warming south of 40S after a couple of hundred years in all experiments, also inferring the Southern Ocean response, though this was not the subject of Schwinger et al. study).
See further comments regarding the simulation of AMOC responses below.

Other points:

Not sure if it is because I am not a native speaker, but I find the title of the manuscript not very intuitive to understand. I would encourage the authors to think about an alternative title.

The title is modified, "The Southern Ocean as the climate's freight train -- driving ongoing global warming under zero-emission scenarios with ACCESS-ESM1.5," so that the message is clearer, even if "freight train" is not.

The use of year 101 as start year for the simulations is confusing. I would suggest to set the nominal start year at year 1 in all tables and figures.

These had been the years from model time of the experiments, which was easier when writing.
But yes, these are easier for the reader by resetting the years as they are presented in the manuscript.

The abbreviations of the simulations is too similar to the abbreviation of ZEC values. For example, the authors use ZEC_200 for the temperature change after 200 years into the ZEC-simulations, and ZEC750 to denote the ZEC simulation with 750 PgC emissions. I would suggest to use a different abbreviation for the simulations.

After some consideration, these experiment terms are still used as is to be something short and clear to help readability of both the text and for use in figures. *"ZeroEmission750"* and *"Branch750"* are long, *"750PgC"* looks too much like a quantity rather than a label, *"A$^{750}$"* doesn't make much sense when B-style experiments are not presented, and *ZE750* is unappealing (and doesn't roll off the tongue as well?). However, a paragraph is added in the Methods now to clarify how symbols and styles are used to describe these ZEC values and experiment labels.

Either in Section 3.2 or 3.3.1, it would be interesting to read something about the role of AMOC changes, which has been identified to play a dominant role in models with strong AMOC decline (Schwinger et al. 2022). Including AMOC strength in Fig 1 could be useful. In Fig 3b it looks like a signal of AMOC decline would be visible in the North Atlantic in the ZEC750 simulation?

Thanks for the comment, Schwinger et al. 2022 and NorESM2-LM results with a strong AMOC signal will be a good comparison.
Regional responses from participating ZECMIP models, including ACCESS-ESM1.5 and NorESM2-LM, are presented now in MacDougall et al. 2022, which highlights the significant regional variability between models and makes special mention of the different impact of AMOC from different models.
Some models show a strong cooling in the North Atlantic that may be associated with a slowdown in the AMOC, and Interestingly, NorESM2-LM is not one of them.
This discrepancy is possibly because the results in MacDougall et al. 2022, are from a lower branch, *ZEC1000,* and earlier in the experiments; ZEC$_{50}$ is before many of the signals in Schwinger et al. 2022 (or our results) become apparent.
Here in the results with ACCESS-ESM1.5, however, the North Atlantic is not standing out particularly in the results presented and we keep our focus on the Southern Ocean which has a strong signal.
The manuscript will be modified to acknowledge and discuss other literature and some of the differences between models, including AMOC.

Section 3.3.1: What is shown in the figures is the global meridional streamfunction not "the overturning". Overturning strength can be visualized through and calculated from the streamfunction. Please correct throughout the manuscript.

The terminology used in the manuscript is clarified as indicated.

Section 4.1: At least the main idea of the slab-model should be described in the main text, such that the reader can understand what the model is intended to do. Might be even easier to move equation A1 into the main text.

(See related comments to "Major Point 2" above)

line 22: I suggest to delete "of the global climate"

The phrase has been left in, it may be useful to the reader who may not be as familiar with the zero-emission commitment term.

line 24: "...the potential budget of carbon emissions permittable without exceeding any agreed thresholds of ``safe'' warming" is very complicated. "remaining carbon budget" has become an established term for this and could be used here.

The suggested terminology is adopted.

line 26: "carbon emission budget" -> "carbon budget"

The suggested terminology is adopted.

line 31: "This conclusion..." this is not a conclusion, it is an assumption.

The suggested terminology is adopted.

line 37: The acronym 1pctCO2 has not been introduced

This sentence with the *1pctCO2*, which is now in a ZECMIP subsection of the methods, is combined with the sentence that followed and has the description of the term.

line 35-41: Much of this paragraph could and should be moved to the methods section.

Moved.

line 70-75: The main issue with trends and biases for the kind of simulations presented here, is the switch from concentration to emission driven configuration. This could be made clear instead of the generic statement in the last sentence of this paragraph.

Thanks, this is a good point, and it is a good argument for the discussions I've heard regarding ZECMIP-style experiments being planned for CMIP7 that branch from a parent experiment that is an emission-based version of the *1pctCO2* experiment.

In the case of ACCESS-ESM1.5, checking output available from the ESGF for the *esm-piControl* that had an interactive carbon cycle, the drift in physical and biogeochemical states is still negligible for the first 300 years. Atmospheric $CO_2$ increases ~1ppm/100y and the magnitude of any trends in the average surface air temperature or sea surface temperature are less than 0.01 degC/100y.

This is now mentioned in the manuscript in this model description subsection of Methods.

line 77-86: The fact that the simulations presented here were run on different computer hardware is a technical detail that is not relevant for the results. This can be a footnote in Table 1 explaining why numerical values are slightly different from previously published results.

These technical comments have been moved from the text to the table caption as suggested.

lines 88-98: This general description of the ZECMIP experiments/protocol should come earlier.

The method section has been rearranged in response to this and other comments.

lines 100-106: Maybe a personal preference, but I don't think it is necessary to provide a summary of subsections at the start of a new section. Good descriptive titles of subsections are enough.

I find the brief high level summaries potentially helpful in communicating the work being presented, and the reader can skim over them easily.

line 109: "...gradient increase evenly" -> "... rate of surface air temperature change" or similar.

This sentence has been broken up and rewritten for clarity.

line 110: The numerical values presented here seem to contradict the values in Table 1? Please clarify.

I can see why this could be the case.

The manuscript is modified to clarify the values in the text are 'overall' values of the rates of change, and that the time series is 'approximately' linear.

Indeed, a quick look over Table shows temperature changes are not linear as $ZEC_{200}$ are not double $ZEC_{100}$, even allowing for uncertainty, as there is a stronger change in the first decades relative to later decades.

line 114-115: This sentence doesn't make sense, please consider rephrasing it.

Rephrased, "Most of the energy entering the climate from the imbalance at the top of the atmosphere (Fig. 1d) is taken up by the ocean of each experiment."

line 130: Unclear what does "the extension of this experiment refer to"? Please clarify.

This had been referring to the new experiments that had been integrated for longer, 300 years rather than 100 years when ACCESS-ESM originally produced results for ZECMIP.

The sentence has been rewritten for clarity…"For instance, while there is an overall global cooling in ZEC750, after 200 years from branching there is some warming in the same latitude band, 40–65◦ S, that stands out more clearly in ZEC1000 (Fig. 2 b and c)."

line 227: TCR is defined at year 70 (not 50) of the 1pctCO2 simulation (at doubling of atmospheric CO2).

Yes, thanks for catching this.

line 238-239: "... more than adequate" I don't think this statement is adequate. The simple model can reproduce certain aspects of the result.

The statement is rephrased to clarify it is only referring to "average temperatures" that are being discussed.

line 256-264: It remains unclear to me what the authors intend to say with this paragraph on tipping points. Please restructure/reword/clarify.

"Tipping points" are often used in describing significant potential impacts of climate change and rightly so. The idea of crossing some threshold of the climate system that triggers a new mechanism or process (e.g. ice sheet collapse, loss of rainforest) that drives the climate to a new state, whether at the local and global scale, is an effective way to get attention.

But after thinking about tipping points for some time, I find the concept is somewhat vague and poorly defined, and potentially missing other important processes.

For a time, our results from the original ZECMIP experiments hinted there may have been a tipping point in our simulations. However, results from our intermediate ZEC branches and then being able to replicate the global results with slab model indicate there is no evidence for any such tipping point driving the global response here.

And yet, here we have a process that indeed affects the climate for centuries, as simulated by ACCESS-ESM1.5, and the process is present in other models as well, albeit with varying impacts globally.

So, the intention here is to suggest that while this Southern Ocean response may not be a tipping point, it is worth being discussed with them…

Editing of the paragraph has been made to expand upon this and make the point clearer.

"While the Southern Ocean and its climate response may not fit an example of a tipping point, its potential to drive ongoing warming with potentially global impacts suggests it should be considered is discussions of regions and processes with potential to drive ongoing changes to the climate system."

line 283-286: There is no "contrast" here this just the different timescale (as the authors note). Please reword these sentences.

These sentences are pointing out how these models that have similar centennial responses but are different on shorter timescales, and sentences have been rephrased for clarity.

References:
Frölicher, T., Winton, M. & Sarmiento, J. Continued global warming after CO2 emissions stoppage. Nature Clim Change 4, 40-44 (2014). https://doi.org/10.1038/nclimate2060
Gillett, N., Arora, V., Zickfeld, K. et al. Ongoing climate change following a complete cessation of carbon dioxide emissions. Nature Geosci 4, 83‚Äì87 (2011). https://doi.org/10.1038/ngeo1047
Schwinger, J., Asaadi, A., Goris, N. et al. Possibility for strong northern hemisphere high-latitude cooling under negative emissions. Nat Commun 13, 1095 (2022). https://doi.org/10.1038/s41467-022-28573-5
Citation: https://doi.org/10.5194/bg-2023-146-RC2

---

## Author Response (AR2)

**Author's Response- bg-2023-146**

M. A. Chamberlain

April 2024.

Thank you to the reviewers for reading again the manuscript and the useful comments.

References have been added and corrected in response to reviewer comments.

Figures indicated (1, 6 and 11) have been checked for colour suitability and find that the results still quite clear with the various forms of colour vision deficiencies.

Further minor corrections have been made following another read through by the authors.

Detailed responses to reviewers follow:

**Comment on bg-2023-146**

Andrew MacDougall

I am satisfied with the changes that the authors have made to the manuscript and believe that that paper is ready for publication.

Thank you and thanks again for your previous review which helped improve the manuscript.

**Comment on bg-2023-146**

Anonymous Referee #2

I find the authors have done a good job in addressing my comments from the first round of review. There are a couple of relatively minor or technical issues that should be addressed before publication in Biogeosciences.

Thank you for reviewing the manuscript again and the useful comments.

Line numbers refer to the tracked changes version of the revised manuscript.

line 1: Wouldn't "Earth system model" be more appropriate instead of "Climate model"?

Yes, this would be more precise.

line 4: "version of" could be deleted.

The phrase is kept here as there are physics-only versions of ACCESS as well.

line 9: I would suggest to delete "or branched".

The phrase is kept here for clarity. Later in the abstract these points where the ESM changes from the 1pctCO2 to zero-emission scenario are referred to as 'branch points' and this is the first occurrence and want the concept to be clear to readers that may be unfamiliar with the running of these climate experiments.

line 32: "of the global climate" and "the amount of" are unnecessary and could be deleted to make this sentence more concise.

The second is deleted to be concise, as suggested, but the first part is kept. The ZEC may not be familiar to all readers and describing it as a property of the 'global climate' could be useful here.

line 38: The authors mean 1.5 (not 1) degree?

Updated.

line 58: "version" should be replaced by "phase".

Updated.

line 64: "50 years into the high ZECMIP branch" maybe better phrased as "50 year after cessation of emissions"?

Suggested phrasing adopted.

line 112: "cumulatively" could be deleted. Also "... after the diagnosed emissions reach 750, ..."

"Cumulatively" is deleted, but keeping "diagnosed emissions" as it refers to the point that these emission budgets must be diagnosed from the *1pctCO2* experiments.  This point is spelt out clearer with the rewriting of this sentence.

line 115: Please replace "into the climate historically" by "for the period xxxx to yyyy" and indicate the years for which the 695 Pg are estimated.

Suggested phrasing adopted.

line 124: "Low (high) ZEC branches..." Please define what you mean by low and high (the reader can guess that "high" means > 1000 Pg(?) but this should be stated here).

With the rewriting from the first review, this paragraph is now somewhat redundant.  The points from this paragraph are now written into the previous and following paragraphs, to make the manuscript more concise and with consideration of the points raised.

line 134: "variability" -> "internal variability".

Suggested phrasing adopted.

line 154-155: I would recommend a more neutral wording, e.g. "... influencing the climate, and an instantaneous transition to net zero carbon emissions would socioeconomically not be feasible".

The wording here was suggested by the first reviewer and I am comfortable with the latest phrasing as the "global cataclysm" is only in the context of the realism, or not, of these *1pctCO2*-to-ZEC transitions and is not belaboured any further.

line 155-158: The statement that "results from ZECMIP experiments are essentially the same" as for other scenario experiments needs clarification (or should be deleted). Which "other" plausible experiments do the authors refer to? Please add a reference to paper that supports this statement.

The language has been softened ("expected to be the same") and a reference is now added to support this statement.

line 179: "the climate" -> "the climate system"

Suggested phrasing adopted.

line 228: Please provide correct citation for "Meehl and IPCC Climate Change 2007"

Yes, thanks, corrected now.

line 258: appears -> appear

Using time series as a singular in this context; "The" has been added to improve readability.

line 263: "However, changes in the North Atlantic and AMOC have been identified..." Maybe better "... have been identified as important features ..."?

Suggested phrasing adopted.

line 335 and 336: "moves to the right" and "they turn left" sound very odd for a description of time series. Please consider rewording.

Modified to "their $CO_2$-temperature trajectories turn left" for clarity.

line 364-365: Please check the logic of this sentence (why "reversed"?). If ZEC is zero, global temperature will be stabilized, but not reversed. Then, in order to stabilize temperatures (in ACCESS) negative emission would be required.

The main point here is that once the Southern Ocean is warming significantly, zero-emissions are not enough and, yes, negative emissions are required, i.e. it is the "global temperature trajectory" (reworded for clarity) that continues to increase, indicating a positive ZEC.
The previous sentence is also rewritten to improve clarity and readability.

line 366: "positions" sounds odd. "thresholds"?

Suggested phrasing adopted.

line 368: Although it might put additional pressure on a system (and eventually push it beyond a tipping threshold), deforestation is not an example of a process related to a tipping point (since it is a deliberate human activity). I guess the authors mean "forest die-back" here?

Yes, "forest die-back" is a better phrase.

lines 369-381: I find the discussion of tipping points lengthy and difficult to read and I would suggest shortening it and making it more concise. I think the main point is that the ongoing warming in the SO isn't a tipping point although the increasing temperatures might cause a crossing of tipping thresholds eventually of components that are not or only poorly represented in ACCESS (Antarctic ice sheet? ocean ecosystems?).

"Climate tipping points" currently receive significant discussion and this paragraph is motivated to point out, in its own small way, that there are other processes that are also significant. I would also add that the Southern Ocean is not a tipping point but still a significant driver of ongoing climate change. The paragraph has been reworked to be more concise and clearer in its message.

line 501: Maybe worth mentioning that ACCESS has a relatively weak (compared to other ESMs) response of AMOC to climate warming?

This and the following sentence are reworked to clarify this point, and that both these AMOC and Southern Ocean responses are plausible climate responses.

line 505: Why is it "reasonable for a climate to be warming"? The low branches traverse the climate states from TCR to ECS without warming, are they "unreasonable"? I would suggest to reword this sentence.
Any trajectory that doesn't cross the bounds of TCR and ECS might be 'reasonable.'
The phrasing has been changed to "not unreasonable" for temperatures to increase with zero-emission scenarios.

---

## Author Response (AR3)

**Author's Response- bg-2023-146**

M. A. Chamberlain
May 2024.

Thank you for accepting our manuscript for publication in Biogeosciences.

In response to the comment from the file validation check (below), the indicated figures have been carefully reviewed. The manuscript results can still be clearly interpretted from the line plots (Figures 1 and 11) even with colour vision deficiencies, and the colour palette has been modified in the top row of panels of Figure 6 where distinctions between regions became ambiguous with some forms of colour blindness.  See below for a detailed response.

The indicated comment read,
"Regarding the figures 1, 6, 11: please ensure that the colour schemes used in your maps and charts allow readers with colour vision deficiencies to correctly interpret your findings. Please check your figures using the Coblis – Color Blindness Simulator (https://www.color-blindness.com/coblis-color-blindness-simulator/) and revise the colour schemes accordingly."

In the case of Figure 1 which shows several lines from various experiments, for the anomalous trichromatic conditions (deuteranomaly, protanomaly and tritanomaly, with prevalences of 2.7%, 0.66% and 0.01% respectively in the total population) the line colours are all still distinct. For red-green dichromatic views (protanopia and deuteranopia, 0.59% and 0.56%) there is a loss of distinction between two of the lines shown.  However, the context of these two lines are quite different and it is straight forward to determine which experiment is represented. One line is the control experiment that is constant on average with time, while the second is the ZEC experiment furthest from the control experiment. For the blue-deficient dichromatic view (tritanopia, 0.015%), different pairs of lines become less distinct, however again, context readily allows a reader to associate the different lines and experiments.

In the case of Figure 11, another series of line plot panels, the lines are distinct for all conditions of colour blindness.

Figure 6 contains multiple panels, including line plots (with the same colour scheme as Figure 1, discussed above), panels with centred-diverging red-blue palettes that are distinct for all colour blindness conditions, and panels on the top row with the absolute-sequential values of salinity.  The colours of this top row were still distinct for anomalous trichromatic conditions, but there was some loss of distinction between upper ocean water at low latitudes and waters with lower salinity at ~ 50°S for dichromatic vision.  These panels in the top row are included primarily for context and are not critical to the findings. However, these panels have been modified in the resubmitted manuscript here with a viridis palette so that it is clear even with colour vision deficiencies.

Regards,
Matt Chamberlain